# Id2 GABAergic interneurons comprise a neglected fourth major group of cortical inhibitory cells

Robert Machold[1], Shlomo Dellal[1†], Manuel Valero[1†‡], Hector Zurita[1], Ilya Kruglikov[1], John Hongyu Meng[1,2], Jessica L Hanson[1§], Yoshiko Hashikawa[1], Benjamin Schuman[1], György Buzsáki[1,3], Bernardo Rudy[1,3,4]*

[1]Neuroscience Institute, New York University Grossman School of Medicine, New York, United States; [2]Center for Neural Science, New York University, New York, United States; [3]Department of Neuroscience and Physiology, New York University Grossman School of Medicine, New York, United States; [4]Department of Anesthesiology, Perioperative Care and Pain Medicine, New York University Grossman School of Medicine, New York, United States

*For correspondence:
Bernardo.Rudy@nyulangone.org

†These authors contributed equally to this work

Present address: ‡Hospital del Mar Medical Research Institute, Barcelona, Spain; §Department of Integrative Physiology and the Institute for Behavioral Genetics, University of Colorado Boulder, Boulder, United States

Competing interest: The authors declare that no competing interests exist.

**Abstract** Cortical GABAergic interneurons (INs) represent a diverse population of mainly locally projecting cells that provide specialized forms of inhibition to pyramidal neurons and other INs. Most recent work on INs has focused on subtypes distinguished by expression of Parvalbumin (PV), Somatostatin (SST), or Vasoactive Intestinal Peptide (VIP). However, a fourth group that includes neurogliaform cells (NGFCs) has been less well characterized due to a lack of genetic tools. Here, we show that these INs can be accessed experimentally using intersectional genetics with the gene *Id2*. We find that outside of layer 1 (L1), the majority of Id2 INs are NGFCs that express high levels of neuropeptide Y (NPY) and exhibit a late-spiking firing pattern, with extensive local connectivity. While much sparser, non-NGFC Id2 INs had more variable properties, with most cells corresponding to a diverse group of INs that strongly expresses the neuropeptide CCK. In vivo, using silicon probe recordings, we observed several distinguishing aspects of NGFC activity, including a strong rebound in activity immediately following the cortical down state during NREM sleep. Our study provides insights into IN diversity and NGFC distribution and properties, and outlines an intersectional genetics approach for further study of this underappreciated group of INs.

## Editor's evaluation

The authors provide a detailed and convincing characterization of a diverse group of cortical interneurons that express the marker Id2, and so can be labeled in Id2-creERT2 mice, but which do not express markers for the previously identified main groups of interneurons: Pvalb, Sst and VIP. This multimodal in vivo and in vitro characterization will be extremely valuable to neuroscientists wishing to study this group of interneurons, which includes previously identified neurogliaform cells and CCK basket cells. in vivo results suggest that these neurons show rebound firing after being inhibited during NREM sleep.

## Introduction

The neocortex is comprised of two primary neuronal classes: pyramidal cells that release the excitatory neurotransmitter glutamate, and interneurons that release the inhibitory neurotransmitter GABA. Pyramidal neurons project their axons locally and distally to form extended circuit ensembles across

cortical and subcortical areas, whereas GABAergic interneurons (INs) typically project their axons locally to regulate the excitability of nearby pyramidal cells and other INs. Pyramidal neurons share a common dendritic structure characterized by an apical dendrite that typically extends up towards the pial surface with extensive arborization in cortical layer 1 (L1), a cell sparse region that receives diverse long-range cortical and thalamic axonal projections (*Garcia-Munoz and Arbuthnott, 2015*; *D'Souza and Burkhalter, 2017*; *Ibrahim et al., 2020*; *Schuman et al., 2021*). In combination with its basal dendrite and somatic inputs, the pyramidal neuron thus has the ability to distinguish and integrate inputs from bottom-up and top-down sources in real-time, a key feature that enables dynamic cortical activity and adaptability during the animal's interactions with the external world (*Schuman et al., 2021*; *Aru et al., 2020*).

While the morphological diversity of IN dendrites and axons was noted over a century ago by Ramón y Cajal, recent work has begun to elucidate how this specialization of IN subtypes allows for compartment-specific inhibition across the entire pyramidal neuron (*Tremblay et al., 2016*; *Feldmeyer et al., 2018*; *Huang and Paul, 2019*; *Fishell and Kepecs, 2020*; *Jiang et al., 2013*; *Jiang et al., 2015*). The discovery of molecular markers in tandem with electrophysical and morphological approaches has enabled the delineation of several major IN groups and subtypes, each with distinctive roles in the cortical circuit (*Tremblay et al., 2016*; *Taniguchi et al., 2011*; *Pfeffer et al., 2013*; *Tasic et al., 2016*; *Paul et al., 2017*; *Tasic et al., 2018*; *Gouwens et al., 2020*; *Yao et al., 2021*). INs that express the $Ca^{2+}$ binding protein parvalbumin (PV), which account for ~40% of the GABAergic INs in neocortex, exhibit fast-spiking properties, with most projecting their axons to form perisomatic baskets on nearby pyramidal neuron soma (*Hu et al., 2014*). In layer 4 of sensory cortical areas, these PV INs form a canonical circuit in which they provide feedforward inhibition of excitatory neurons receiving core thalamic inputs (*Bruno and Simons, 2002*; *Cruikshank et al., 2007*). A distinct but less abundant subtype of PV INs are the chandelier cells, whose axons form cartridge-like inputs on the pyramidal neuron axon initial segment, thereby enabling control and coordination of the axonal output of pyramidal neuron ensembles (*Inan and Anderson, 2014*; *Lu et al., 2017*; *Gallo et al., 2020*). Somatostatin (SST) expressing interneurons (accounting for ~30% of the INs) represent a second major group of INs, with the majority characterized by an ascending axonal arbor that targets the distal apical dendrites of pyramidal cells (SST Martinotti cells) (*Tremblay et al., 2016*; *Riedemann, 2019*). A third major group of INs is distinguished by the expression of the neuropeptide vasoactive intestinal peptide (VIP); these interneurons account for ~12% of the INs in the neocortex and include cells that preferentially target other interneurons, particularly SST INs, and thus act to disinhibit the local circuit (*Lee et al., 2013*; *Pi et al., 2013*; *Keller et al., 2020*; *Kullander and Topolnik, 2021*). The vast majority of recent studies exploring the roles of INs in cortical function have focused on these three groups, in no small part due to the establishment of genetic tools in rodents for targeting and manipulating the activity of each group (*Taniguchi et al., 2011*; *Madisen et al., 2015*; *He et al., 2016*; *Daigle et al., 2018*). This body of work has revealed how IN diversity facilitates greater flexibility and computational power in cortical circuits.

In addition to the three main IN groups described above, there remains an additional population of GABAergic interneurons (about 18% of the INs in the neocortex) that have been much less studied. This IN population is particularly enriched in superficial layers of the neocortex, and includes ~90% of the INs in layer 1 (*Tremblay et al., 2016*; *Schuman et al., 2019*) as well as neurogliaform cells (NGFCs) in all layers and CCK basket cells (CCK BCs), two IN types previously identified in the neocortex that do not express PV, SST or VIP (*Kisvárday et al., 1990*; *Kawaguchi and Kubota, 1996*; *Galarreta et al., 2004*; *Bodor et al., 2005*). NGFCs are an interneuron type that exhibits several remarkable properties, including an extremely high degree of output connectivity to other cells (pyramidal neurons and other INs) and the ability to induce postsynaptic GABA-B responses following a single action potential (*Tamás et al., 2003*; *Oláh et al., 2007*; *Oláh et al., 2009*; *Overstreet-Wadiche and McBain, 2015*). CCK BCs are a rare species in the neocortex whose synapses on pyramidal neurons exhibit endocannabinoid mediated depolarization induced suppression of inhibition (DSI; *Bodor et al., 2005*; *Galarreta et al., 2008*).

Cortical GABAergic INs are specified during embryogenesis in the medial and caudal ganglionic eminences of the ventral telencephalon (MGE and CGE, respectively; *Wamsley and Fishell, 2017*; *Hu et al., 2017*; *Lim et al., 2018*), with a fraction of non-MGE derived INs originating from the preoptic area (POA; *Gelman et al., 2009*). All cortical PV and SST INs arise from Nkx2.1+progenitors within the

MGE (*Xu et al., 2008*), and express Lhx6 during tangential migration and in the adult (*Liodis et al., 2007*). Cortical CGE derived INs, i.e., the non-PV, non-SST subtypes labeled in the Htr3a(BAC)-EGFP mouse line (the 5HT3aR INs; *Lee et al., 2010*) can be divided into two primary populations: VIP- and non-VIP-expressing (*Tremblay et al., 2016*; *Rudy et al., 2011*; *Vucurovic et al., 2010*). While transgenic targeting approaches for PV, SST, and VIP INs have been developed and widely implemented (*Taniguchi et al., 2011*; *He et al., 2016*), accessing non-VIP CGE (or 5HT3aR) INs – a fourth major IN group – has been challenging due to a lack of genetic strategies to target this population, and therefore the IN subtypes present in this population are not well characterized.

Here, we show that Id2 is a marker for the INs in the neocortex that do not express PV, SST or VIP, including the non-VIP cells in L1, NGFCs in L2-6 and a variety of CCK INs. Using intersectional genetic strategies to label and optogenetically stimulate Id2 INs, we find that the vast majority of Id2 INs in L2-6 are NPY-expressing NGFCs that exhibit the late-spiking firing pattern and broad connectivity characteristic of these cells. In contrast, the much rarer non-NGFC Id2 INs display diverse firing patterns and DSI but sparse local connectivity. Combining opto-tagging with silicon probe recordings, we observe several intriguing features of NGFC in vivo behavior, such as a robust rebound in activity immediately following the cortical down states during NREM sleep. Thus, our work furthers the study of IN diversity and enables genetic access to a group of INs that despite their potent GABAergic output have remained somewhat overlooked to date.

## Results

We examined cortical single-cell transcriptome data (scRNAseq) from the Allen Institute (*Tasic et al., 2016*) and observed that expression of the Id2 gene was highly enriched in the INs in their collection that do not express PV, SST, or VIP (*Figure 1A*; *Mayer et al., 2018*). The Id2 IN population includes cells that express markers known to be present in non-VIP CGE INs such as Reelin (*Miyoshi et al., 2010*), and NPY (*Kubota et al., 2011b*), as well as the more recently identified markers NDNF, Lamp5 and Sncg (*Tasic et al., 2016*). Updated and expanded cortical and hippocampal scRNAseq data from the Allen Institute (*Yao et al., 2021*; *Allen Institute for Brain Science, 2023*) confirms that Id2 is a useful marker for studying non-PV, SST, or VIP INs (*Figure 1—figure supplement 1*). To develop a transgenic approach to target Id2 INs, we took advantage of an existing Id2-CreER driver line (*Rawlins et al., 2009*) that expresses a tamoxifen-inducible form of Cre recombinase (*Metzger et al., 1995*). Since Id2 is also expressed in pyramidal and non-neuronal cell types in the cortex, we crossed the Id2-CreER driver with a pan-IN Flp driver (Dlx5/6-Flpe; *Miyoshi et al., 2010*) such that only Id2 INs would be targeted with intersectional reporters (*Figure 1B*). In combination with the Cre and Flp dependent tdTomato reporter Ai65 (*Madisen et al., 2015*), this intersectional genetics approach yielded labeling of INs in all cortical layers of the somatosensory cortex barrel field (S1BF) following tamoxifen administration (3–4 doses between P21 and P35; see Materials and methods) to activate the CreER (*Figure 1C*).

### Histological analysis of Id2 Ins

To determine the proportion of Id2 vs. non-Id2 INs, we utilized the intersectional/subtractive reporter FLTG (*Plummer et al., 2015*) where Flp activity results in tdTomato expression, but Flp + Cre results in EGFP labeling (*Figure 1D*). Quantification of Id2 INs (green) and non-Id2 INs (red) in the S1BF of well-labeled brains (see Materials and methods) yielded an overall fraction of 18% (680/3812 cells, n=4 brains), with the majority of Id2 INs located in superficial layers (L1-3; *Figure 1E–F*). This proportion and distribution was consistent with our previous estimates for non-VIP 5HT3aR or CGE INs (*Tremblay et al., 2016*), indicating that this genetic strategy was efficiently targeting this population. We observed a similar overall proportion of Id2 INs in other sensory areas (e.g. V1), consistent with a recent comprehensive survey of interneuron subtype distribution (*Yao et al., 2021*), as well as in a higher order cortical area, the prelimbic cortex (18%; *Figure 1—figure supplement 2*).

To assess any potential overlap between Id2 INs and PV, SST, or VIP INs, we performed intersectional genetic labeling experiments with the respective Flp drivers that cover those populations. Both PV and SST INs originate from Nkx2.1+progenitors (i.e. MGE lineages; *Xu et al., 2008*), and are efficiently labeled using an Nkx2.1-Flpo driver line (*He et al., 2016*). However, there is also a population of neurogliaform INs (NGFCs) that arise from an Nkx2.1+lineage, the vast majority of which migrate

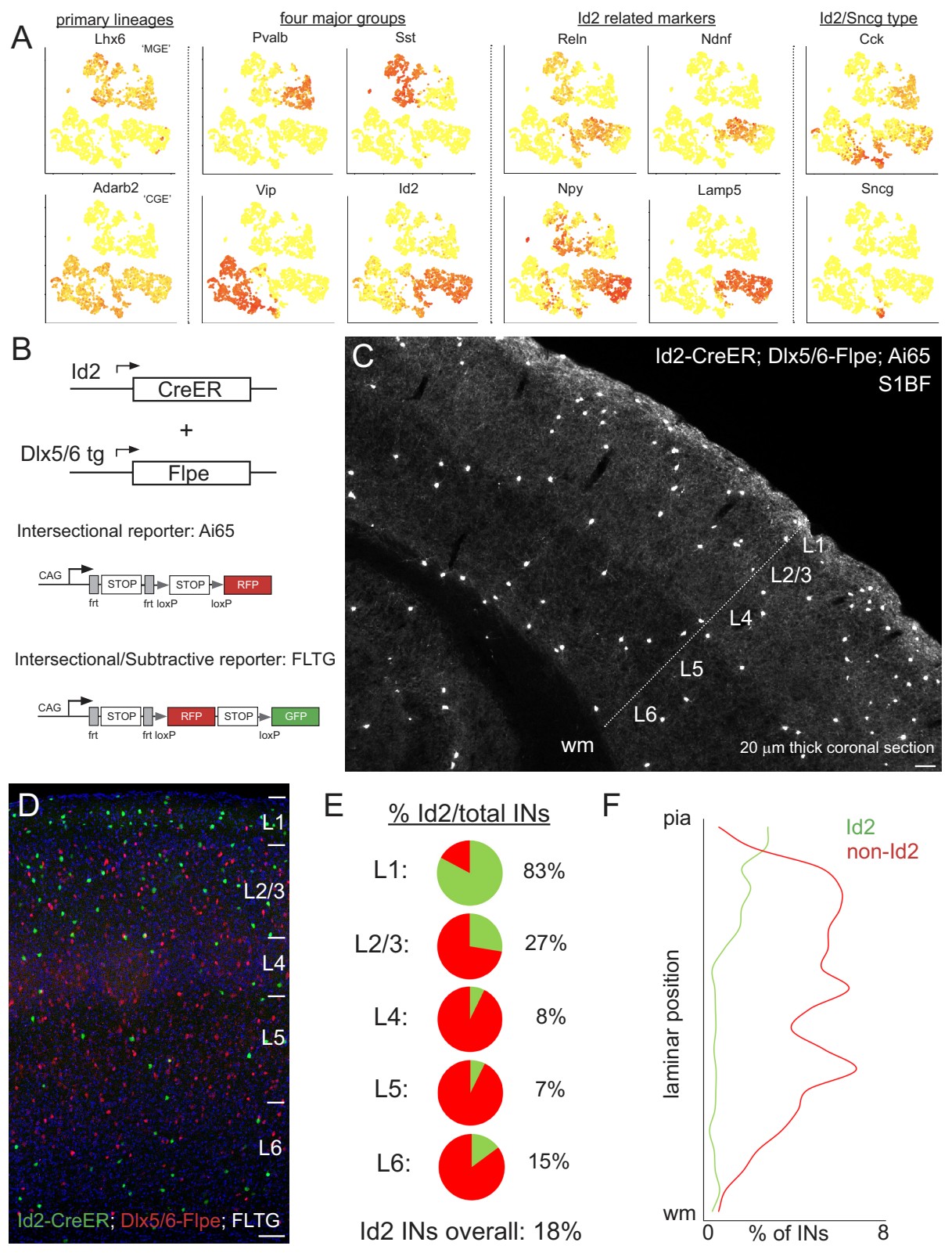

**Figure 1.** Id2 expression delineates a fourth major group of INs. (**A**) Public scRNAseq data (from *Tasic et al., 2016*; comprised of ~3000 INs purified from the visual cortex, with transcriptome diversity represented as tSNE plots; see *Mayer et al., 2018*) underlying the hypothesis that the non-VIP CGE-derived IN population can be distinguished by the expression of *Id2*. Primary IN lineages originating from the medial (MGE) and caudal (CGE) ganglionic eminences express *Lhx6* and *Adarb2*, respectively. To a first approximation, MGE-derived INs are comprised of parvalbumin (*Pvalb*) and

*Figure 1 continued on next page*

*Figure 1 continued*

somatostatin (*Sst*) subtypes, whereas CGE-derived INs consist of vasoactive intestinal peptide (*Vip*) and inhibitor of DNA binding 2 (*Id2*) subtypes. Expression of the marker genes Reelin (*Reln*), neuron-derived neurotrophic factor (*Ndnf*), neuropeptide Y (*Npy*) and cholecystokinin (*Cck*) is evident within the Id2 population, which roughly corresponds to the lysosomal associated membrane protein 5 (*Lamp5*) plus synuclein-γ (*Sncg*) IN categories in *Tasic et al., 2016*. (**B**) Intersectional genetic strategies for targeting Id2 INs. The Id2-CreER knock in driver line combined with the Dlx5/6-Flpe transgenic line allows for labeling of Id2 INs with tdTomato when crossed with the Ai65 intersectional reporter (or the channelrhodopsin CatCh when crossed with Ai80). The use of the intersectional/subtractive reporter FLTG enables the dual color labeling of Id2 and non-Id2 INs. (**C**) Image of a coronal cryosection (20 μm thick) of an Id2-CreER; Dlx5/6-Flpe; Ai65 labeled brain (**P30**) reveals the distribution of Id2 INs throughout the layers of the cortical S1 barrel field (S1BF). (**D**) Image of a coronal field in S1BF of Id2-CreER; Dlx5/6-Flpe; FLTG labeled brain with Id2 INs in green and non-Id2 INs in red. (**E**) Proportion of Id2 vs. non-Id2 INs across the layers of S1BF determined from cell counts of the cross in (**D**). Id2 INs comprise 18% of the overall IN population (3812 cells counted across 4 brains; see also *Figure 1—source data 1*). (**F**) Distribution of Id2 and non-Id2 IN somas across S1BF lamina. Scale bars in panels C and D represent 100 μm.

The online version of this article includes the following source data and figure supplement(s) for figure 1:

**Source data 1.** Id2 vs. non-Id2 cell counts in S1BF.

**Figure supplement 1.** IN scRNAseq gene expression heat maps.

**Figure supplement 1—source data 1.** mRNA counts in Id2 cells.

**Figure supplement 2.** Prevalence and distribution of Id2 INs in prelimbic cortex.

**Figure supplement 2—source data 1.** Id2 vs. non-Id2 cell counts in PrL.

**Figure supplement 3.** Id2 INs show little overlap with Nkx2.1-lineage or VIP cells.

**Figure supplement 4.** Methodological considerations for assessing gene expression (the cautionary tale of CCK).

**Figure supplement 5.** Comparison of 5HT3aR lines in S1BF.

to the hippocampus (*Tricoire et al., 2010*). A small fraction of these MGE-derived NGFCs is located in the neocortex; these are the sparse deep layer Lamp5/Lhx6 cells described and characterized previously by us using intersectional genetics with Id2 (Id2-CreER; Nkx2.1-Flpo; Ai65) (*Krienen et al., 2020*; *Valero et al., 2021*). Id2/Nkx2.1 cells did not show any overlap with PV or SST, and were mostly localized in L6 (*Valero et al., 2021*; *Figure 1—figure supplement 3*). To examine possible overlap with VIP populations, we generated Id2-CreER; VIP-Flpo; Ai65 animals and assessed the degree of IN labeling. Here too, we observed very few labeled cells in S1BF, with most being located along the L1/2 border (*Figure 1—figure supplement 3*). While Id2/VIP cells were extremely sparse in S1BF, they were somewhat more prevalent in the prelimbic cortex along the L1/2 border (*Figure 1—figure supplement 3*). Thus, we find that the vast majority of Id2 INs (>95%) are distinct from PV, SST or VIP populations, and in the cortex, almost all originate from non-Nkx2.1 lineages (i.e., from the CGE/POA).

In superficial layers (L1-3), expression of neuropeptide Y (NPY) has been utilized successfully as a proxy to identify NGFCs (*Schuman et al., 2019*; *Kubota et al., 2011b*; *Chittajallu et al., 2013*; *Neske et al., 2015*). We evaluated the overlap between Id2 INs and NPY by performing fluorescence ISH (FISH) for tdTomato and NPY on tissue cryosections from Id2-CreER; Dlx5/6; Ai65 brains (S1BF; *Figure 2A–B*). The proportion of Id2/NPY cells vs. the total Id2 labeled population in S1BF ranged from 29% in L1 to 82% in L2/3, 95% in L4-5, and 61% in L6, with a combined proportion in L2-5 of 87% (638 cells counted, sections from three brains), suggesting that in contrast to L1, where Id2 INs consist of several subtypes (*Schuman et al., 2019*), the vast majority of Id2 cells outside of L1 are NGFCs. Within L1, our findings are consistent with our previous analysis that about 30% of L1 INs are NPY-expressing NGFCs, with the remainder of the Id2 population being comprised of canopy cells (NDNF/non-NPY) and α7 cells (*Schuman et al., 2019*). We further confirmed that most Id2 INs in L2-6 are NPY-expressing throughout different sensory cortical areas by generating Id2-CreER; Dlx5/6-Flpe; Ai65; NPY-hrGFP animals and comparing labeling in S1BF, V1 and A1 (*Figure 2—figure supplement 1*).

To explore the nature of the non-NPY Id2 neurons in L2-6, we examined the overlap between Id2 labeled cells and relatively high levels of CCK expression as the latter is a proxy for a sparse population of INs that include the large CCK basket cells (*Freund and Katona, 2007*; *Freund, 2003*; *Kawaguchi and Kubota, 1997*). Using IHC for CCK (*Figure 2C–D*), overall, in L2-6 of S1BF we observed that 15% of Id2 labeled INs exhibited strong CCK expression (165/1123 cells, sections from three brains), with Id2/CCK cells being most abundant in L2 (14% of Id2 cells) and L6 (25%). No overlap

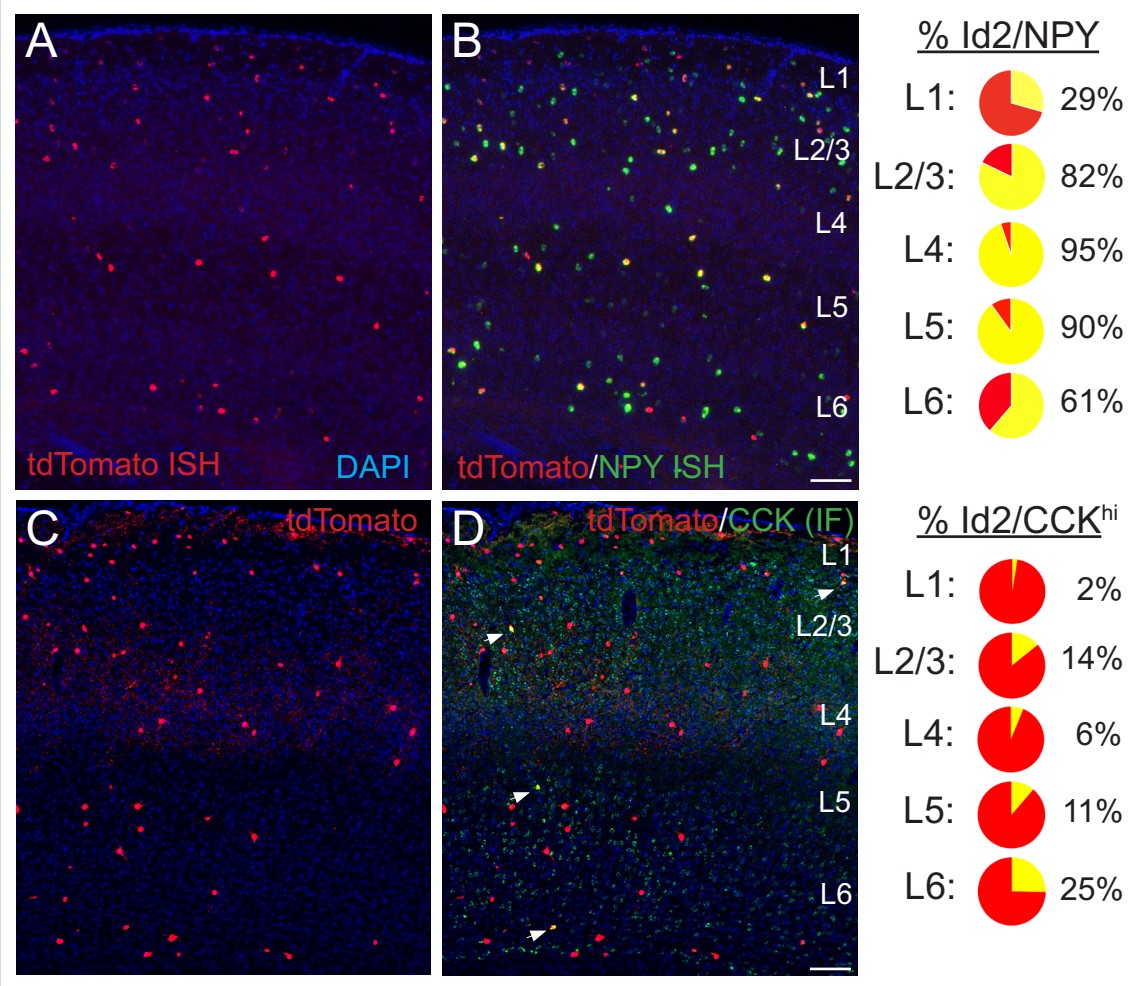

**Figure 2.** Histological analysis of Id2 INs in S1BF. (**A–B**) Assessment of *Npy* expression in Id2 INs (Id2-CreER; Dlx5/6-Flpe; Ai65). Fluorescent in situ hybridization (FISH) with cRNA probes for *tdTomato* (red; panel A) and *Npy* (green; both red and green channels shown in panel B) mRNAs on brain cryosections (20 μm thick) reveals that outside of L1, the vast majority of Id2 INs express *Npy* (L2-5: 87%). The percentages of Id2/*Npy* cells (yellow) in each layer are represented as pie charts on the right. (**C–D**) Assessment of CCK expression in Id2 INs (Id2-CreER; Dlx5/6-Flpe; Ai65). Immunofluorescent histochemistry (IHC) for tdTomato (red; panel A) and CCK (green) reveals that a fraction (15%) of Id2 cells in L2-6 express high levels of CCK (CCK$^{hi}$; white arrowheads), with the highest proportion of Id2/CCK cells observed in L2 and L6. The percentages of Id2/CCK cells (yellow) in each layer are represented as pie charts on the right. Scale bars in panels B and D represent 100 μm. See also *Figure 2—source data 1*.

The online version of this article includes the following source data and figure supplement(s) for figure 2:

**Source data 1.** Id2 NPY CCK cell counts in S1BF.

**Figure supplement 1.** Id2 INs and NPY expression in S1BF, V1, and A1.

was observed between strong CCK IHC and NPY-hrGFP expression, consistent with previous work (*Kubota and Kawaguchi, 1997*) and the current Allen scRNAseq data (*Figure 1—figure supplement 1*; *Yao et al., 2021*). Thus, a significant fraction of the Id2/non-NPY population in cortical layers outside of L1 appear to be CCK +IN subtypes (~80% in L2/3), such that Id2/NPY and Id2/CCK populations account for nearly all of the Id2 INs in L2-5.

## Electrophysiological and morphological properties of Id2 Ins

Our FISH analysis (*Figure 2A–B*) suggested that the majority (87%) of Id2 INs in L2-5 are NPY-expressing NGFCs. To test this hypothesis, we characterized the firing properties of Id2 INs in L2-5 of S1BF in acute brain slices. The cells were filled with biocytin during electrophysiological recording for post-hoc morphological analysis. NGFCs in L2/3 and L1 have been shown to have a late-spiking (LS) firing pattern characterized by delayed firing following a slow ramp depolarization when depolarized

with step current injections to near-threshold membrane potentials (*Tremblay et al., 2016*; *Schuman et al., 2019*; *Tamás et al., 2003*; *Oláh et al., 2009*; *Overstreet-Wadiche and McBain, 2015*). NGFCs in these layers were also found to fire spike trains with little spike frequency adaptation, or even spike frequency acceleration, during depolarizations close to threshold, and mild adaptation during suprathreshold depolarizations. Action potentials in NGFCs have a large AHP and a slow ADP (*Schuman et al., 2019*; *Tamás et al., 2003*; *Oláh et al., 2009*; *Hestrin and Armstrong, 1996*; *Wozny and Williams, 2011*; *Kawaguchi, 1995*). We found that in a sample of randomly recorded Id2 cells (n=76), the large majority in L2-5 had a LS firing pattern (36/45; 80% in L2/3 and 28/31; 90% in L4-L5; *Figure 3*). These proportions are similar to the proportion of Id2/NPY cells (*Figure 2*). The Id2 LS cells also showed weak spike frequency adaptation during near threshold membrane depolarizations and had other electrophysiological properties typical of NGFCs (*Table 1*). In paired recordings of L2/3 LS Id2 cells and PCs, we observed a very high connection probability (70%; 19/27 pairs tested), consistent with previous observations and the hypothesis that NGFCs mediate volume transmission of GABA (*Oláh et al., 2009*). Finally, the decay of the IPSC elicited by NGFCs is known to be prolonged compared to that mediated by other INs (*Tamás et al., 2003*; *Oláh et al., 2007*). Consistent with this, we found that the decay rate (80 to 20% of peak) of the IPSC in LS-PC pairs was significantly slower than that in non-LS-PC pairs (LS-PC: 51±8ms, n=6 pairs; non-LS-PC: 16±3ms, n=5 pairs; p=0.005, two-tailed unpaired t-test).

Morphologically, the LS cells in L2-5 had the characteristic features of NGFCs with short multipolar dendrites emanating around the cell body and a significantly larger, dense axonal arbor surrounding the cell body and dendritic arbor (*Figure 3A–B* and *Figure 3—figure supplement 1*; n=15), consistent with previous descriptions of NGFC in these layers across several neocortical areas (*Tamás et al., 2003*; *Oláh et al., 2007*; *Oláh et al., 2009*; *Overstreet-Wadiche and McBain, 2015*; *Kawaguchi, 1995*). NGFCs in L1 resemble NGFCs in L2-5 in having short dendrites and dense axonal arborization, but their axonal arbor extends for longer horizontal distances, spanning several columns (*Jiang et al., 2013*; *Jiang et al., 2015*; *Schuman et al., 2019*; *Hestrin and Armstrong, 1996*; *Kubota et al., 2011a*; *Zhou and Hablitz, 1996*) and have been called "elongated neurogliaform cells" (*Jiang et al., 2013*; *Jiang et al., 2015*). We found that in the LS cells located in deeper layers, the axonal arbor was also asymmetrical, but often extended in the vertical (columnar) direction. These differences between NGFCs across cortical layers are also evident in the NGFC morphologies illustrated in a recent patch-seq study (*Gouwens et al., 2020*).

The remaining Id2 cells (~20% in L2/3 and ~10% in L4-5) had either an irregular or a bursting firing pattern during low to moderate depolarizations (*Figure 3C–D and F*, and *Table 1*) and narrower action potentials (*Figure 3E*). We also observed that a small minority of the non-LS Id2 cells in L2/3 (~15%) were likely α7 cells, a L1 Id2 IN subtype typically localized in the deeper part of L1, which has a bursting firing pattern that can be distinguished from other L2/3 bursting cells by the properties of the burst, significantly lower input resistance, and morphology (*Figure 3—figure supplement 1*; *Schuman et al., 2019*). The histological analysis showed that the majority (~80% in L2/3) of the non-NPY Id2 cells strongly express the neuropeptide CCK (*Figure 2*). CCK is expressed at varying levels by many types of cortical neurons, including GABAergic and glutamatergic cells; however, it is particularly high in a group of GABAergic interneurons known as CCK-CB1R interneurons or CCK basket cells that have been described in the hippocampus and L2/3 of the neocortex (*Galarreta et al., 2004*; *Galarreta et al., 2008*; *Freund and Katona, 2007*; *Glickfeld and Scanziani, 2006*; *De May and Ali, 2013*). Although much less abundant than other GABAergic interneurons in the neocortex, they are of special interest because of their unique synaptic properties (*Glickfeld and Scanziani, 2006*; *Hefft and Jonas, 2005*; *Armstrong and Soltesz, 2012*). In particular, these cells express CB1 cannabinoid receptors in their presynaptic terminals that mediate a phenomenon known as depolarization-induced suppression of inhibition (DSI) (*Pitler and Alger, 1992*; *Wagner and Alger, 1996*). Depolarization of a connected pyramidal cell leads to the release of endocannabinoids from the pyramidal cell that then bind to CB1 receptors located on the presynaptic terminals of the CCK basket cells, leading to suppression of GABA release (*Wilson et al., 2001*; *Wilson and Nicoll, 2001*; *Ohno-Shosaku et al., 2001*).

To test the hypothesis that non-LS Id2 neurons include CCK basket cells, we first examined the effect of depolarization of a pyramidal cell on the PSP generated on L2/3 PCs by light stimulation of Id2 neurons expressing the engineered channelrhodopsin variant CatCh (Id2-CreER; Dlx5/6-Flpe;

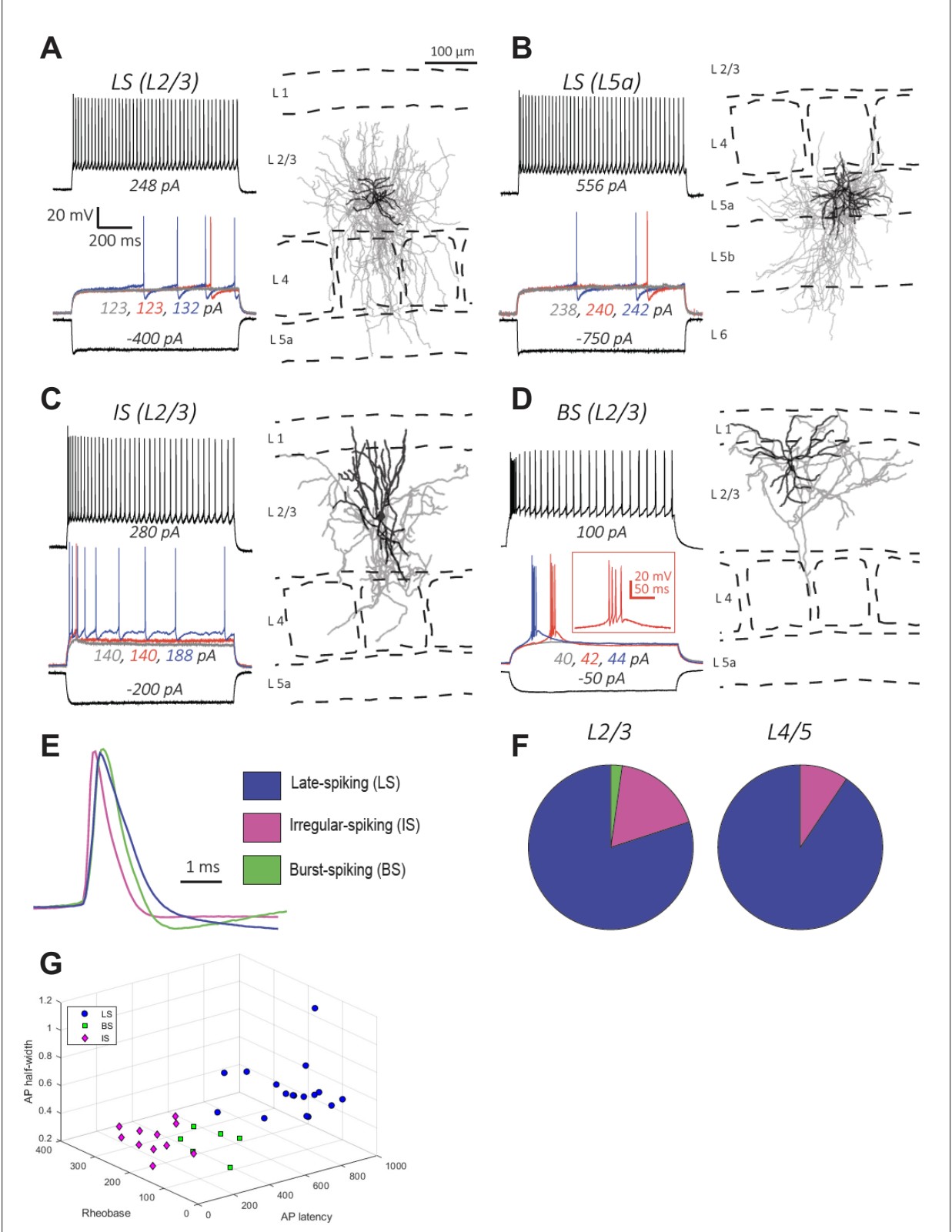

**Figure 3.** Electrophysiology and morphology of Id2 INs in S1BF. (**A–D**) Four examples of the types of cells encountered in this survey and their respective locations in the cortical column: (**A–B**) Late-spiking (LS), (**C**) Irregular-spiking (IS), and (**D**) Burst-spiking (BS). For electrophysiological characterization, cells were injected with current to bring them to –70 mV. Each voltage trace shows the response of the cell to a 1 s long current injection at the indicated levels which are: (i) a negative current to hyperpolarize the cell to –100 mV [black-lower trace], (ii) current steps just below

*Figure 3 continued on next page*

*Figure 3 continued*

[grey trace], at [blue trace], and just above rheobase [red trace], and (iii) current at roughly double the rheobase [black-upper trace]. To the right of each cell's voltage traces is the morphological reconstruction from biocytin fills of the same recorded cell with the cortical layers and barrel fields indicated with dashed lines. Dendrites are shown in black and axons are shown in gray. (**E**) Overlay of first action potential waveforms at rheobase from the three described cell types. (**F**) Proportions of the three main cell types encountered in L2/3 and L4/5. For L2/3, 45 randomly selected cells were patched and characterized electrophysiologically. The proportions of the cell types were: 80% LS, 18% IS, and 2% BS. For L4/5, 31 randomly selected cells were characterized, and the proportions of the cell types were: 90% LS, 10% IS, and 0% BS. (**G**) 3D plot of LS, BS, and IS Id2 INs resolved by AP half-width, rheobase, and AP latency.

The online version of this article includes the following figure supplement(s) for figure 3:

**Figure supplement 1.** Additional morphologies of Id2 INs.

Ai80; *Figure 4A–C*; *Daigle et al., 2018*; *Kleinlogel et al., 2011*). Depolarization of the PC reduced the synaptic response by ~20%, and this reduction was prevented by application of the CB1 receptor antagonist AM251 (*Figure 4B–C*) suggesting that some of the presynaptic cells innervating the cell had undergone DSI. In paired recordings of L2/3 LS Id2 cells and PCs, we found that the synaptic response elicited by NGFCs did not undergo DSI (n=9; *Figure 4E*). In contrast, in paired recordings of non-LS Id2 cells (of 46 nonLS-PC pairs tested, only 9 (20%) were connected), we observed DSI in three out of the five pairs tested (*Figure 4E*). Of the cells showing DSI, two were irregular spiking and one was bursting; of the two cells that did not show DSI, one was burst spiking and the other was a putative α7 cell. These results indicate that the partial suppression of GABA release observed when the whole population of Id2 cells is stimulated is occurring primarily on the terminals of the non-LS Id2

**Table 1.** Electrophysiological properties of Id2 INs in S1BF.

Values are reported as Mean ± SEM. Significance was tested for each group against LS in L2/3 using the Mann-Whitney test, * denotes significance between 0.01 and 0.05, ** denotes significance between 0.001 and 0.01, and *** denotes significance <0.001. Significance was not assessed for the first spike latency as cells were categorized on that basis. For some cells, not all of the parameters were assessed, so the n values are the minimum number of cells used to calculate all of the parameters. See Materials and Methods section for how these parameters were obtained. See also *Table 1—source data 1*.

| Electrophysiological property | LS in L2/3 (n≥15) | LS in L4-5 (n≥5) | IS in L2/3 (n≥5) | BS in L2/3 (n≥5) |
|---|---|---|---|---|
| First AP latency (ms) | 805±22 | 827±32 | 53±4 | 143±37 |
| Input Resistance (MΩ) | 188±7 | 130±11*** | 199±32 | 301±32*** |
| Membrane τ (ms) | 12±0.5 | 11±0.8 | 13±1.2 | 17±1.4** |
| AP half-width (ms) | 0.70±0.02 | 0.63±0.08 | 0.48±0.04*** | 0.57±0.05* |
| AP threshold (mV) | 35±1 | 33±2 | 38±2 | 41±2* |
| AP rise slope (mV/ms) | 283±15 | 367±61 | 404±60 | 322±46 |
| AP fall slope (mV/ms) | 84±4 | 108±20 | 150±19*** | 135±29* |
| AHP amplitude (mV) | 14.4±0.5 | 12.9±1.1 | 9.9±0.9*** | 6.6±1.3*** |
| Rheobase (pA) | 139±14 | 205±44 | 155±18 | 51±7*** |
| Firing rate (Hz) (at 2 x rheobase) | 38±3 | 29±4 | 40±6 | 31±7 |
| Firing Regularity (at 2 x rheobase) | 0.053±0.004 | 0.050±0.008 | 0.112±0.029** | 0.262±0.124** |
| Adaptation Index (near rheobase) | 1.14±0.08 | 1.06±0.11 | - | - |
| Adaptation Index (at 2 x rheobase) | 0.76±0.03 | 0.72±0.03 | 0.26±0.05*** | 0.24±0.14** |

The online version of this article includes the following source data for table 1:

**Source data 1.** Electrophysiological parameters source data.

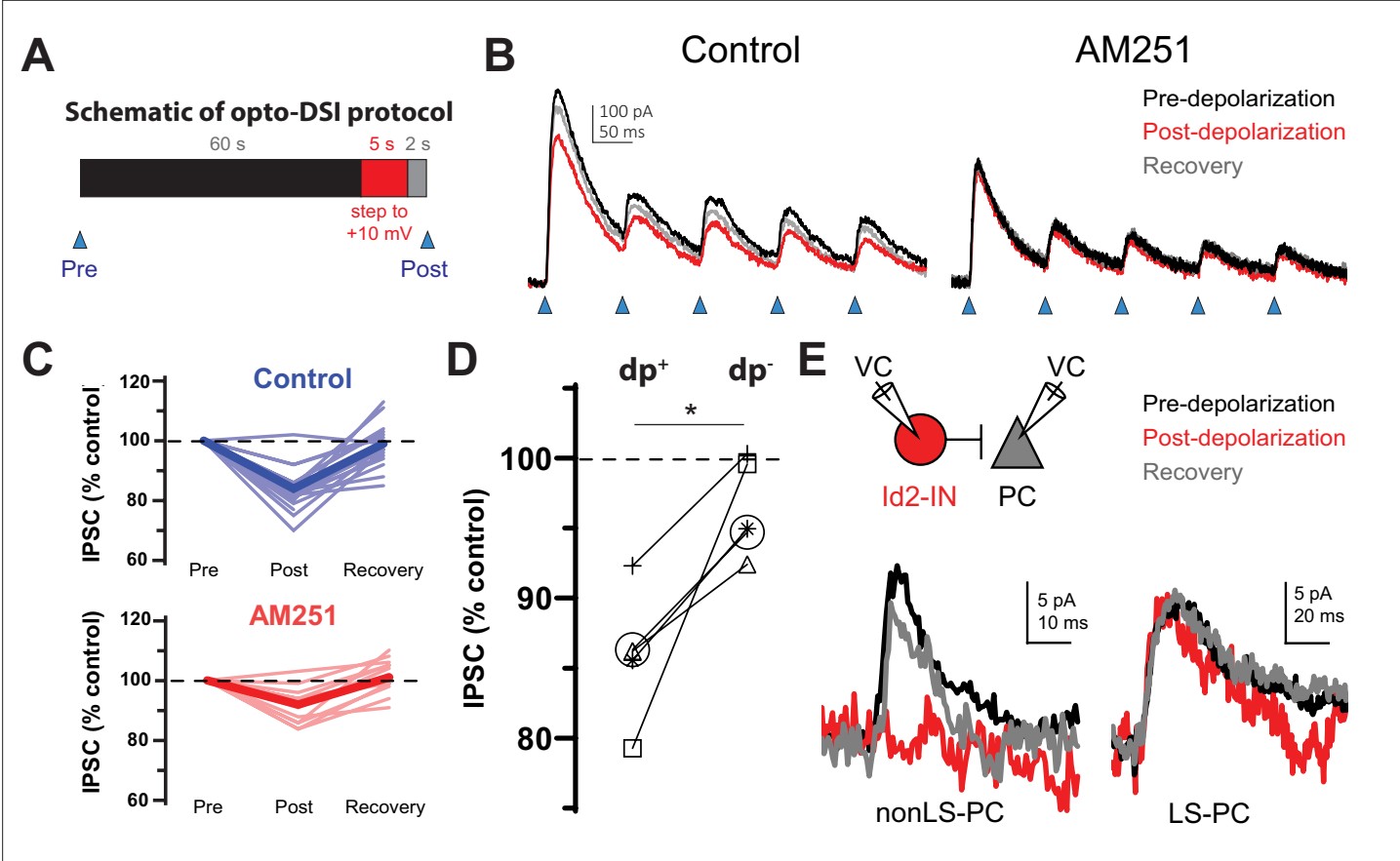

**Figure 4.** The non-LS Id2 population exhibits DSI. (**A**) Protocol used for revealing depolarization-induced suppression of inhibition (DSI). DSI was first assessed via optogenetics in L2/3 pyramidal cells (PCs) in the following manner: PCs were recorded in voltage-clamp mode in cortical slices from Id2-CreER; Dlx5/6-Flpe; Ai80 animals, in which the channelrhodopsin CatCh is widely expressed in the Id2 IN population. Control IPSCs were elicited by light pulses of 2ms in duration. Then, after a 60 s interval, the PC was stepped to +10 mV for 5 s, and after a 2 s recovery, another IPSC was elicited in the same manner. (**B**) Sample traces from optogenetic experiments. In the drug-free condition (left panel), the peak current decreases slightly but noticeably and recovers by the next stim 60 s later. However, in the presence of the CB1 receptor antagonist AM251 (10 μM) [right panel], this decrease was not observed. (**C**) Summary data from all cells tested as in (**B**). Thin, faint lines are the averages for each cell across three or more repetitions. Dark, solid lines are the averages across all cells. The average decrease in IPSC amplitude in the control condition was 16 ± 2% (n=18 cells, N=8 animals). Two-tailed paired t-tests of Post vs. Pre and Recovery vs. Post were significant (p=1e-6, p=6e-6). The average decrease in IPSC amplitude in the AM251 condition was 8 ± 1% (n=11 cells, N=4 animals). Also, the average decrease post-depolarization in Control vs. AM251 was significant by a two-tailed unpaired t-test (p=0.002). (**D**) A subset of five cells was tested in which the DSI protocol was carried out with (dp+) or without (dp-) a depolarization to +10 mV to control for variability or rundown of the synaptic current. Trials that incorporated the depolarization step exhibited significantly greater decreases in IPSC amplitude than those that lacked it (p=0.014, two-tailed paired t-test). (**E**) DSI was also detected in nonLS-PC pairs but not in LS-PC pairs. Recording configuration and sample current traces from two PCs: a post-synaptic PC to a nonLS cell (left) and a post-synaptic PC to an LS cell (right). Current traces represent a DSI experiment. Several probe stimuli were given to obtain a baseline average (black trace), then the depolarization step to +10 mV was delivered to the PC, and 1–2 s later, the PC was probed again (red trace). The cell was then probed several times afterwards to assess recovery (grey trace). The mouse lines used for these experiments were Id2-CreER; Dlx5/6-Flpe; Ai65 or the same with the addition of an NPY-hrGFP allele to facilitate the identification of nonLS cells. Out of 46 nonLS-PC pairs tested, only 9 (20%) were connected (compared to 24% for CB$_1$IS→P in *Galarreta et al., 2008*). Out of 27 LS-PC pairs tested, 19 (70%) were connected. DSI was assessed in 9 LS-PC pairs and not observed in any of them, whereas out of 5 nonLS-PC pairs tested for DSI, 3 pairs exhibited it. See also *Figure 4—source data 1*.

The online version of this article includes the following source data for figure 4:

**Source data 1.** DSI source data.

cells. Consistent with this, the proportion of the GABAergic response elicited by light activation of Id2 cells that was suppressed by depolarization of the post-synaptic PC is similar to the representation of non-LS cells in the L2/3 Id2 population. These observations, together with the molecular analysis, suggest that the majority of the non-LS Id2 cells are CCK basket cells. The morphology of IS and BS non-LS Id2 cells was similar, but quite distinct from the NGFCs, with significantly longer dendrites

and a less symmetrical and much less dense axonal arborization (*Figure 3C–D* and *Figure 3—figure supplement 1*), consistent with the published morphologies of CCK basket cells in the neocortex (*Galarreta et al., 2004*; *De May and Ali, 2013*).

## In vivo properties of Id2 INs

Although NGFCs are a substantial population of GABAergic INs, particularly in the superficial layers of the cortex (where they are slightly more abundant than SST INs), the lack of genetic strategies to study these cells has limited the characterization of their in vivo properties. Nevertheless, given their broad output connectivity, these neurons are likely to contribute significantly to inhibition throughout the cortical column. We utilized our intersectional genetics approach to express CatCh in Id2 INs (Id2-CreER; Dlx5/6-Flpe; Ai80; also see *Figure 4*) in order to optogenetically identify and stimulate these cells in multichannel silicon probe recordings in the posterior parietal cortex and V1 of freely moving and behaving animals (*Figure 5A*). Single units were isolated (n=571 units from 5 mice) and separated into putative pyramidal cells (PC), narrow waveform (NW) INs and wide waveform (WW) INs (*Petersen et al., 2021*). Id2 INs expressing CatCh were distinguished from other cells adjacent to the probes by measuring their firing rates in response to light stimulation (*Figure 5B*). Applying stringent filtering criteria (see Methods), we selected 11 Id2 INs that exhibited robustly increased firing following light stimulation for further analysis (*Figure 5B–C*). We then characterized the firing properties of these Id2 INs identified in vivo in relation to PCs and other INs. These Id2 INs exhibited WW characteristics, and were readily distinguishable from NW cells, previously identified as PV fast-spiking INs and some SST INs (*Valero et al., 2021*; *Figure 5B and D*). Consistent with this, these 11 Id2 INs had wider auto-correlograms than other PC and NW INs, similar to the overall WW group (*Figure 5E*). These results indicate that the previously described WW group (*Petersen et al., 2021*) includes an unknown proportion of Id2 INs.

In addition to the 11 WW Id2 INs described above, we also observed two light responsive Id2 INs with narrower waveform properties, although these cells did not pass the selection criteria for further analysis. We estimate that the majority of Id2 INs that could be identified by optotagging in our silicon probe experiments were located in deeper layers (~L3-5), where the proportion of Id2 cells that are NGFC is >90% (*Figure 2*). Thus, we tentatively conclude that the 11 Id2 WW INs we selected for analysis are NGFCs, and that the Id2 cells exhibiting narrower waveform properties are likely to be non-NGFC CCK +types.

The firing rates of PC, NW and WW neurons varied across behavioral states, typically showing lower activity during sleep (*Figure 5F*). In contrast, putative Id2 NGFC INs exhibited relatively consistent firing rates across all conditions, with moderately increased activity during REM sleep (*Figure 5F*). During NREM sleep, the cortex interleaves periods of high spiking activity (up-states) with transient silences (down-states) (*Valero et al., 2021*; *Steriade et al., 1993*). To gain insights into the nature of the firing dynamics of Id2 INs during sleep, we aligned the activity of all recorded neurons to the peak of the down-state (*Figure 5G*). Most of the neurons decreased their activity during down states, and steadily recovered their baseline activity after the down-state (post-down; *Figure 5H*). Putative Id2 NGFCs decreased their activity during down states to a similar extent as the other cell groups, but strongly increased their firing above baseline levels immediately following the termination of the down-state (*Figure 5H*, bottom;~4 s.d. above pre-down rate for Id2; 0.2, 0.6 and –0.03 for PC, NW and WW, respectively). This rebound in activity was observed in the majority of the putative Id2 NGFCs (*Figure 5G*, right; 64% of the Id2 NGFCs vs. 7%, 14% and 1% for the PC, NW, and WW interneurons, respectively; $p<10^{-10}$, chi-squared test) and was negatively correlated with the degree of firing suppression during down-states for the Id2 group (*Figure 5I*), suggesting that intrinsic membrane properties of the Id2 INs could be mediating, at least in part, their increased activity after down-states. No other neuron group showed a significant correlation between down and post-down state activity, notwithstanding the considerably bigger sample size (*Figure 5I*). Furthermore, Id2 INs fired at an early position during the up-states sequences (*Figure 5J*), as estimated by the normalized rank order during the detected up-state epochs (see Materials and methods).

Finally, to understand how Id2 IN activity impacts cortical circuit dynamics in vivo, we performed optogenetic stimulation of the Id2 IN population and analyzed which fraction of putative PC, NW, or WW were suppressed (neg mod in *Figure 6A*), activated (pos mod) or unchanged (no mod). Light stimulation resulted in a significant decrease in firing rate for about half of the population of all

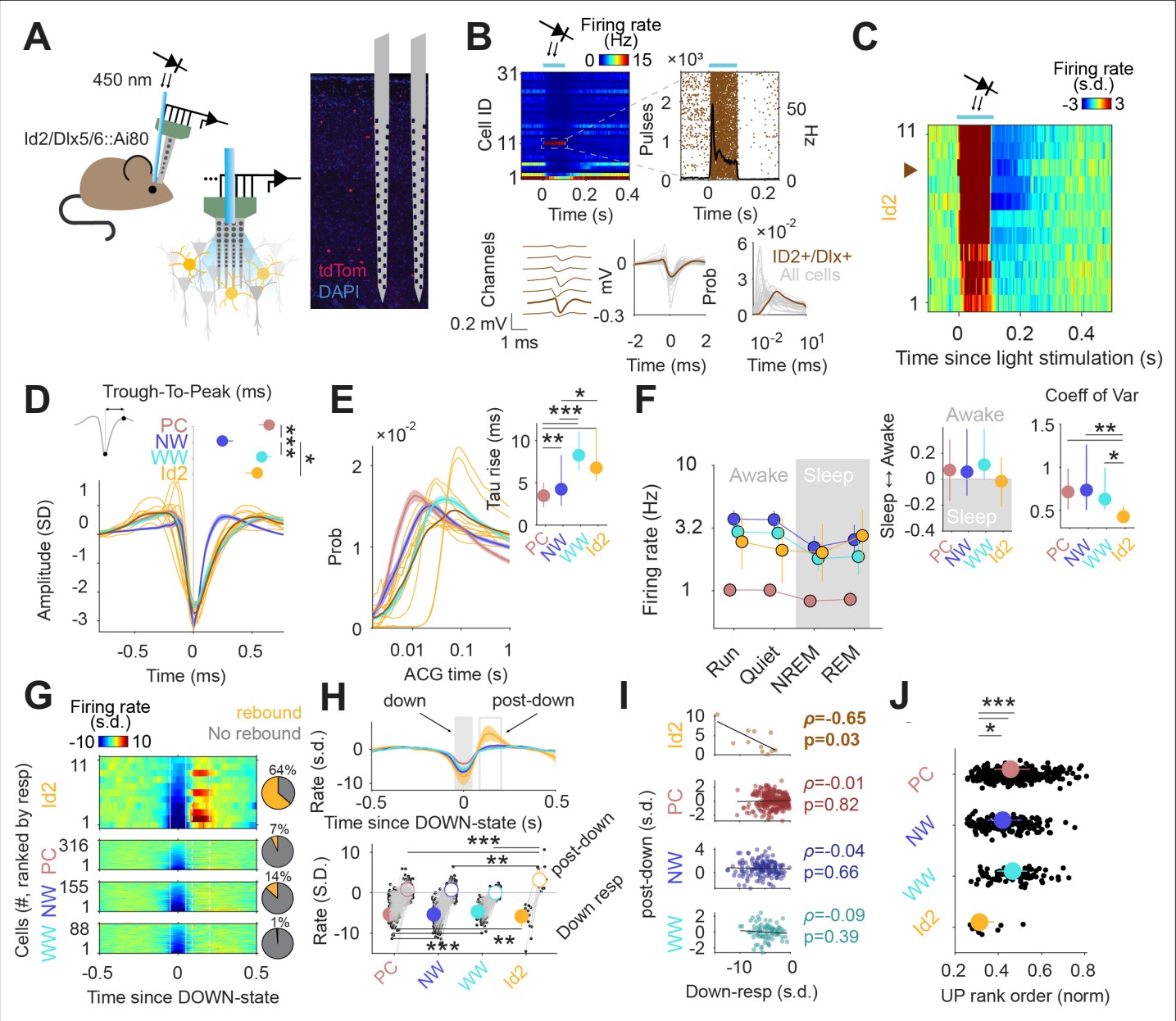

**Figure 5.** Activity of Id2 INs in vivo. (**A**) Schematic of the opto-tagging experiments that were used to identify Id2 INs in freely behaving mice. Combined light fiber-recording probes were implanted in Id2-CreER; Dlx5/6-Flpe; Ai80 mice. (**B**) Peristimulus time histogram (PSTH) from 31 isolated neurons from a single recording session. Note one light-responsive neuron in position 11. The peristimulus raster plot, waveform and auto-correlogram for that light-responsive Id2 IN-light responsive unit are shown in brown. All remaining cells from this session are shown in clear gray for comparison. (**C**) Peristimulus histogram (PSTH) for all isolated units classified as Id2 IN-light responsive neurons (from n=5 mice). (**D**) Average spike waveform (mean ± 95% confidence interval (CI95)) and through-to-peak spike duration (inset; p<10⁻⁶¹, Kruskal-Wallis test) for PCs, NW INs, WW neurons and Id2 INs (n=317 PCs, 155 NW INs and 88 WW neurons). The average waveform from all Id2 IN-light responsive neurons are overlayed in yellow. (**E**) Same as in (**D**) but for the average firing auto-correlogram (ACG) in log scale and time constant of the rising phase of the ACG (inset; p<10⁻²⁴). (**F**) Median firing rates of PCs, NW INs, WW neurons, and light responsive-Id2 INs (Cell type: $F_{3,2261}$ = 151.44, p<10⁻⁸⁹; State: $F_{3,2261}$ = 10.74, p=0.035; Interaction: $F_{9,2261}$ = 0.97, p=0.46). Sleep to Awake index (p=0.051, Kruskal-Wallis test) and coefficient of variance of the firing rate (p=0.034, Kruskal-Wallis test) as insets. Sleep to Awake index was near zero only for the Id2 IN group (PC: Z=2.55, p=0.010; NW: Z=3.16, p=0.002; WW: Z=0.08, p=0.98; one-sample Wilcoxon signed rank test). (**G**) Peri-down-state Z-scored firing histogram for light-responsive-Id2 INs (top), and all PCs, NW INs, WW neurons (middle to bottom), as ranked according to their magnitude of event response. Solid lines delimit the ±50 ms window used to estimate the unit responses during down-states. Dashed lines delimit the +90 ms to +200 ms window used to estimate the post-down-state responses. Pie charts show the fraction of units exhibiting an early increase in firing rate during the post-down epoch above 2 standard deviations (s.d.). (**H**) Temporal dynamics of the Z-scored peri-down-state responses across all groups (top; mean ± IC95). Contrary to what was observed in PCs, NW INs and WW neurons, Id2 neurons significantly

*Figure 5 continued on next page*

*Figure 5 continued*

increased their firing rate above baseline levels during the post-down epoch (post-down: $p<10^{-8}$; down: $p<10^{-15}$). (**I**) The magnitude of the post-down-state response as a function of the activity during the down-state events. Only Id2 neurons show a significant correlation (Spearman correlation). (**J**) Group differences of the average rank position during up-state epochs ($P<10^{-5}$, Kruskal-Wallis test). Id2 INs lead the up-state sequences.

considered cell types (*Figure 6A*), consistent with the broad direct inhibition expected from NGFCs. PCs in the supra-granular layers were more frequently suppressed than deeper layer PCs, while both NW and WW INs showed the opposite trend across lamina. Interestingly, we also observed a slow positive activity modulation of a subset of INs, and to a lesser extent PCs, following stimulation of Id2 INs (*Figure 6A*; darker quantiles in *Figure 6B–C*). Positively modulated responses consistently lagged the negative responses (*Figure 6B–D*), suggesting an indirect disinhibitory mechanism for the positive responses. Overall, we find that the recruitment of Id2 INs leads to a wide range of response dynamics throughout the cortical column, including both inhibitory and to a lesser extent, relatively slower disinhibitory effects.

## Discussion

The use of molecular markers to classify subpopulations of cortical GABAergic interneurons (INs) has provided deep insights into IN diversity (*Tremblay et al., 2016*; *Kawaguchi and Kubota, 1997*; *Petilla Interneuron Nomenclature et al., 2008*; *De Felipe et al., 2013*), and has expanded in recent years with the advent of single-cell RNA sequencing methods to profile IN transcriptomes (*Tasic et al., 2016*; *Tasic et al., 2018*; *Yao et al., 2021*; *Zeisel et al., 2015*). Identification of differentially expressed genes has enhanced our ability to design and implement genetic strategies to label and experimentally manipulate defined populations of INs in order to understand their specialized functions in cortical circuitry. However, several critical issues have emerged from this body of work that require consideration. Differentially expressed genes that exhibit distinct labeling patterns as assessed by immunohistochemistry (IHC) or in situ hybridization (ISH) may also be expressed at lower levels in broader cell populations that escape detection with those methods. In transgenic animals expressing recombinases such as Cre, labeling strategies that utilize viral or genetically encoded Cre-dependent reporters typically reveal these additional populations due to the lower expression threshold required for labeling with these genetic methods compared to IHC or ISH. For example, expression of the marker CCK as assessed by IHC or ISH has been used to identify subsets of INs with relatively high

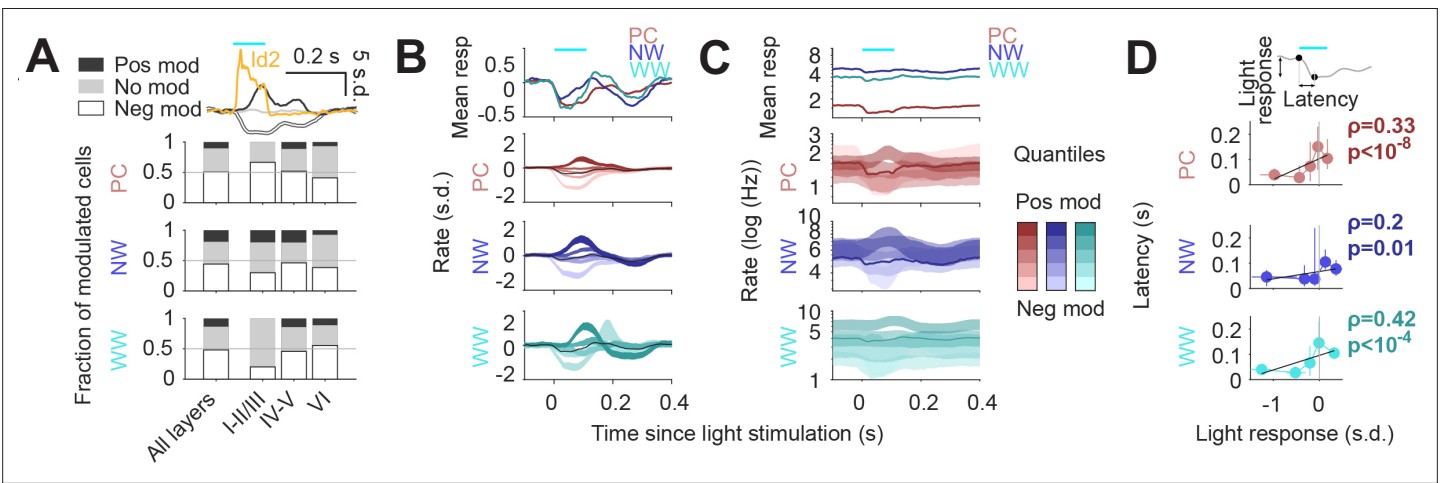

**Figure 6.** Circuit effects of Id2 IN stimulation in vivo. (**A**) Fraction of negatively modulated (neg mod), positively modulated (pos mod), and unmodulated neurons (no mod) in the PC, NW and WW groups across cortical layers. Average peristimulus histogram for all three modes of response (pos mod, neg mod, and no mod) is shown in the top inset. (**B–C**) Peristimulus histograms for PCs, NW INs, and WW neurons, each separated into 5 quantiles based on their + 0 ms to +100 ms average firing rate response (light to dark shades). Changes in firing rates are shown as s.d. in (**B**) and Hz (log10 scale) in (**C**). (**D**) Peak latency for the light-evoked responses in all neuron groups as a function of the +0 ms to +100 ms rate response. Positively modulated responses peaked consistently later than negative responses (Spearman correlation).

levels of CCK, but the use of CCK-Cre with a genetic reporter results in much more extensive labeling of INs (*Figure 1—figure supplement 4*) as well as pyramidal cells.

Another important consideration is the developmental trajectory of marker gene expression, in that the adult expression pattern for a particular marker may be much more restricted than its cumulative expression. This appears to be the case for the marker 5HT3aR (Htr3a), where expression of EGFP from a BAC transgenic mouse line labels virtually *all* non-PV or SST INs (i.e. all the CGE-derived INs), a finding that led to the conclusion that PV, SST and 5HT3aR INs account for all the INs in the neocortex (*Lee et al., 2010*; *Rudy et al., 2011*). We compared labeling efficiencies obtained with a knock in Htr3a-Flpo line (*Schuman et al., 2019*) with that of an Htr3a(BAC)-Cre line (*Gerfen et al., 2013*) and the Htr3a(BAC)-EGFP line used by *Lee et al., 2010*, and found that two copies of the Htr3a-Flpo driver could largely recapitulate the cell labeling observed with the higher copy number BAC transgenic lines (*Figure 1—figure supplement 5*). This result, along with ISH images of Htr3a expression at embryonic and adult ages (e.g. Allen Brain Atlas) is consistent with broad but transient Htr3a expression in non-PV/non-SST INs that then becomes restricted to a subpopulation of these INs (*Prönneke et al., 2020*).

Use of BAC transgenics has turned out to be more complicated than expected since each transgenic founder harbors its own positional and copy number variation. The Htr3a(BAC)-EGFP founder line was specifically chosen over others for high levels of EGFP expression. In hindsight, this likely contributed to its pan-CGE IN expression pattern, whereas endogenous Htr3a expression appears to be broad but transient in most CGE INs following their specification, being largely restricted to VIP/CCK and Sncg IN subtypes at adult ages. In contrast to the Htr3a(BAC)-EGFP line, the Htr3a(BAC)-Cre founder was selected based on its cumulative recombination pattern with the cre-dependent tdTomato reporter Ai9, and likely has a much lower copy number or genomic insertion location that results in its expression being closer to the endogenous Htr3a gene. The Htr3a-Flpo line is a single copy knock in whose cumulative labeling with the tdTomato reporter Ai65F is less efficient overall than that seen with either BAC transgenic line, most likely due to the lower/transient expression of Flpo in most CGE INs.

Leveraging the scRNAseq resources provided by the Allen Institute (*Tasic et al., 2016*), we identified Id2 as a marker for the vast majority of the INs that do not express PV, SST, or VIP. While Id2 is also expressed in pyramidal neurons and several non-neuronal cell types, we employed intersectional genetics with a pan-IN driver (Dlx5/6-Flpe) to restrict labeling to Id2 INs. This labeling approach relies on the use of an Id2-CreER driver, which has its own pros and cons: tamoxifen induction of cre activity in Id2 cells in P21 or older animals avoids spurious cell labeling arising from developmental Id2 expression, but the efficiency of CreER-mediated recombination can be highly variable from animal to animal depending on the effectiveness of tamoxifen administration. Overall, in well-labeled brains, we found that Id2 INs comprised ~18% of the total IN population in S1BF, which is consistent with our previous estimates for the prevalence of non-PV/non-SST/non-VIP INs (*Tremblay et al., 2016*). In L1, a layer consisting only of GABAergic INs, we previously identified three distinct subpopulations of non-PV/non-SST/non-VIP INs: NGFCs, canopy cells, and α7 cells, which together accounted for ~90% of the neurons in this layer, with the remaining 10% consisting of VIP INs (*Schuman et al., 2019*). Using our Id2 intersectional genetic strategy, we find that 83% of L1 INs are labeled (*Figure 1E*), close to the expected 90% of non-PV/non-SST/non-VIP INs, suggesting that all non-VIP INs in L1 are Id2 cells.

In L2-5, we find that Id2 INs are mostly comprised of NGFCs (~87%), with the remainder consisting of a diverse CCK +IN population. While to date NGFCs have mainly been recorded in superficial layers (L1-3), here we observe that NGFCs are present in all layers, including deep layers, but are enriched in L2/3. On the other hand, CCK +INs are mainly located in superficial L2/3 along the border with L1 and in L6. As previously shown for the 5HT3aR INs (*Tremblay et al., 2016*), compared to L4-5, the proportion of Id2 INs increases in L6, where this group accounts for about 15% of all INs (*Figure 1*). Within the L6 Id2 population, about 60% were NPY+ (putative NGFCs), and about 25% were strongly CCK+, with about 10% unaccounted for (*Figure 2*). Within the Id2 NGFC in L6, a subset originates from the MGE (i.e., Nkx2.1 lineage) and exhibit a striking anti-correlated activity pattern, with peak activity during cortical down-states (*Valero et al., 2021*). We did not observe down-state active cells in our Id2 IN recordings (*Figure 5G*), indicating that NGFCs of CGE origin (the majority of cortical NGFCs) behave in a distinct manner in vivo. Thus, like L1, L6 is a specialized case that merits its own study to elucidate the diverse IN species that reside there.

The massive scale of a recent transcriptomic cell type survey from the Allen Institute (*Yao et al., 2021*) offers an unprecedented look at neocortical and hippocampal IN diversity, and presents the opportunity to assess the expression profile of specific genes across all IN types. Focusing on the 'CGE' branch of interneurons, we find that Id2 expression largely encompasses the 'Lamp5' and 'Sncg' branches of their online dendrogram (*Figure 1—figure supplement 1*). However, it is important to note that neither Lamp5 or Sncg are uniformly expressed across the different subtypes in their respective branches. In the case of Sncg, many of the CCK +subtypes within that branch express low if any *Sncg* mRNA itself (e.g. within the Serpinf1 sub-branch). Furthermore, there is substantial overlap with VIP in the Sncg population overall (~50% of cells), but this VIP/Sncg population does not appear to be well labeled using our Id2 strategy (*Figure 1—figure supplement 3*), likely due to the lower levels of Id2 in these cells (*Figure 1—figure supplement 1*). The recent development of new drivers for Lamp5 (Lamp5-Flpo; Jax #037340) and Sncg (Sncg-Flpo; Jax#034424) (*Dudok et al., 2021*) will enable new transgenic targeting approaches for these respective IN populations, but the breadth and specificity of labeling achieved with these new driver lines remain to be determined. Future work will hopefully resolve how this relatively sparse but remarkably diverse CCK +IN population functions within the cortex, and whether some of these cells might relate to primate-specific CCK +IN types such as the rosehip cell (*Boldog et al., 2018*).

The genetic strategies described here have also facilitated the in vivo recording of NGFCs. Outside of L1, where NGFCs express NDNF, a gene that has been used to facilitate the identification and in vivo recording of these neurons (*Cohen-Kashi Malina et al., 2021*; *Ibrahim et al., 2021*; *Abs et al., 2018*; *Hay et al., 2021*), recording of NGFCs has been limited to blind recordings followed by post-hoc identification (*Sakalar et al., 2022*; *Fuentealba et al., 2010*; *Klausberger et al., 2003*), a very low yield approach. Here, we combined optogenetic labeling via Id2 intersectional genetics with silicon probe recordings to identify putative NGFCs in vivo (*Figure 5*). We identified 11 Id2 INs located approximately in L3-5 where the vast majority (>90%) of Id2 INs are NGFCs, as defined by histology and electrophysiology (*Figures 2–4*). We found that these putative NGFCs have a number of unique properties, including relatively constant firing rates throughout different brain states and a strong rebound in activity immediately following the down-state during NREM sleep. We also found that optogenetic stimulation of Id2 INs produced broad inhibition of PCs and other INs, with roughly half of all cells across cortical layers exhibiting a decrease in activity. The notable rebound in NGFC activity following the down-state, together with the extensive output connectivity of these neurons, suggests an important role for this form of GABAergic inhibition in setting the stage for new incoming information during bouts of cortical activity. Future work will certainly continue to elucidate the function of this distinctive GABAergic cell type in regulating cortical circuitry.

# Materials and methods

**Key resources table**

| Reagent type (species) or resource | Designation | Source or reference | Identifiers | Additional information |
|---|---|---|---|---|
| Genetic reagent (mouse) | Id2-CreER | Jax | 016222 | |
| Genetic reagent (mouse) | Dlx5/6-Flpe | Jax | 010815 | |
| Genetic reagent (mouse) | NPY-hrGFP | Jax | 006417 | |
| Genetic reagent (mouse) | Nkx2.1-Flpo | Jax | 028577 | |
| Genetic reagent (mouse) | VIP-Flpo | Jax | 028578 | |
| Genetic reagent (mouse) | Ai65 | Jax | 021875 | |
| Genetic reagent (mouse) | Ai80 | Jax | 025109 | |
| Genetic reagent (mouse) | FLTG | Jax | 026932 | |
| Antibody | Rabbit polyclonal anti-CCK | Frontier Institute | Af350 | 1:500 dilution |
| Antibody | Donkey polyclonal anti-rabbit AlexaFluor-488 | Invitrogen | A-21206 | 1:1000 dilution |

*Continued on next page*

*Continued*

| Reagent type (species) or resource | Designation | Source or reference | Identifiers | Additional information |
|---|---|---|---|---|
| Sequence-based reagent | Cre primer (fwd) | Invitrogen | custom | caacgagtgatgaggttcgca |
| Sequence-based reagent | Cre primer (rev) | Invitrogen | custom | cgccgcataaccagtgaaaca |
| Sequence-based reagent | Flpe primer (fwd) | Invitrogen | custom | tctttagcgcaaggggtagga |
| Sequence-based reagent | Flpe primer (rev) | Invitrogen | custom | aagcacgcttatcgctccaa |
| Sequence-based reagent | Flpo primer (fwd) | Invitrogen | custom | ccacattcatcaactgcggc |
| Sequence-based reagent | Flpo primer (rev) | Invitrogen | custom | gggccgttcttgatagcgaa |
| Sequence-based reagent | NPY-hrGFP primer (fwd) | Invitrogen | custom | atgtggacggggcagaagatc |
| Sequence-based reagent | NPY-hrGFP primer (rev) | Invitrogen | custom | gtgcggttgccgtactgga |
| Sequence-based reagent | NPY cRNA antisense probe primer (fwd) | Invitrogen | custom | tcacagaggcacccagagc |
| Sequence-based reagent | NPY cRNA antisense probe primer (rev +T7) | Invitrogen | custom | attaatacgactcactatag cggagtccagcctagtggtg |
| Sequence-based reagent | tdTomato cRNA antisense probe primer (fwd) | Invitrogen | custom | atcaaagagttcatgcgcttc |
| Sequence-based reagent | tdTomato cRNA antisense probe primer (rev +T7) | Invitrogen | custom | cattaatacgactcactataggg ttccacgatggtgtagtcctc |
| Chemical compound, drug | tamoxifen | Sigma | T5648 | Prepare 20 mg/ml stock in corn oil |

## Mice

The following primary lines of mice were used to generate the compound transgenic crosses described in this paper: Id2-CreER (Jax #016222) (*Rawlins et al., 2009*), Dlx5/6-Flpe (Jax #010815) (*Miyoshi et al., 2010*), NPY-hrGFP (Jax #006417) (*van den Pol et al., 2009*), Nkx2.1-Flpo (Jax #028577) (*He et al., 2016*), VIP-Flpo (Jax #028578) (*He et al., 2016*), Ai65 (Jax #021875) (*Madisen et al., 2015*), Ai80 (Jax #025109) (*Daigle et al., 2018*), and FLTG (Jax #026932) (*Plummer et al., 2015*). The reporter lines Ai65, Ai80 and FLTG were maintained as homozygous stocks prior to crossing with compound heterozygous driver lines (e.g. Id2-CreER; Dlx5/6-Flpe X Ai65 homozygous), with the resulting pups genotyped for Cre and Flpe to obtain experimental animals (e.g. Id2-CreER; Dlx5/6-Flpe; Ai65). To generate Id2-CreER; Dlx5/6-Flpe; Ai65; NPY-hrGFP animals, a stock of Ai65 (homozygous); NPY-hrGFP (heterozygous) animals was established and then crossed with Id2-CreER; Dlx5/6-Flpe breeders. Genotyping was performed on genomic DNA extracted from pup toe clippings using the following primers (sequences are 5' to 3'): Cre (fwd primer: caacgagtgatgaggttcgca, rev primer: cgccgcataacc agtgaaaca, product = 304 bp), Flpe (fwd primer: tctttagcgcaaggggtagga, rev primer: aagcacgcttat cgctccaa, product = 312 bp), Flpo (fwd primer: ccacattcatcaactgcggc, rev primer: gggccgttcttgatag cgaa, product = 355 bp), and NPY-hrGFP (fwd primer: atgtggacggggcagaagatc, rev primer: gtgc ggttgccgtactgga, product = 400 bp). To activate CreER, a stock solution of tamoxifen (Sigma T5648; dissolved in corn oil at 20 mg/ml on a shaker at 55 °C for several hours) was administered to P21-P35 animals by oral gavage (4 mg/20 g mouse) 3–4 x over 5–7 days. The efficacy of tamoxifen induction of CreER mediated reporter activation was assessed by histology (see below); well labeled brains typically exhibited >50 labeled cells in a 20 µm thick tissue section of S1BF encompassing 4–5 barrels (~1300 µm x 1700 µm field). All animals were group housed in an SPF barrier facility on a 12 hr light/dark cycle, with unlimited access to food and water. All experiments were performed in accordance with protocols approved by the Department of Comparative Medicine at the NYU Grossman School of Medicine.

## Histology

To obtain brain tissue for histology, animals (P40-P60) were transcardially perfused with 4% paraformaldehyde/PBS (diluted from a 32% stock; Electron Microscopy Sciences cat#15714) and the brain dissected, with a post-dissection fix period of 0–16 hr. For the preparation of thin cryosections,

brains were equilibrated in 30% sucrose/PBS overnight before freezing in mounting medium (Tissue-Plus O.C.T. compound; Scigen 4583). For the quantification of Id2 vs non-Id2 interneurons in *Figure 1*, vibratome sections (50 µm) were generated following perfusion/PBS washes using a Leica vibratome (VT1000S). Brain cryosections (20 µm) were generated using a Leica CM3050 cryostat, collected on glass slides (Shandon ColorFrost Plus; ThermoScientific cat#9991013), and stored at –20 °C. For immunohistochemistry (IHC), tissue sections were washed in PBS, blocked for 1 hr at room temperature with PBS/0.1% Triton-X 100/2% normal donkey serum, incubated overnight at 4 °C with primary antibody (rabbit anti-CCK; Frontier Institute cat#Af350; 1:500 dilution) in blocking solution, washed in PBS, incubated 1 hr at RT with secondary antibodies (for CCK IHC: donkey anti-rabbit AlexaFluor-488; Invitrogen cat#A-21206; 1:1000 dilution). Fluorescence in situ hybridization for NPY and tdTomato (*Figure 2A–B*) was performed as described previously (*Schuman et al., 2019*), with the following primers used to amplify templates for labeled cRNA antisense probe synthesis: NPY fwd primer: 5'-tcacagaggcacccagagc-3'; NPY rev primer (w/T7 sequence): 5'-attaatacgactcactatagcggagtccagcctagtggtg-3'; tdTomato fwd: 5'-atcaaagagttcatgc gcttc-3'; tdTomato rev primer (w/T7 sequence): 5'-cattaatacgactcactatagggttccacgatggtgtagtcctc -3'.

## Slice preparation

Adult transgenic mice of either sex (postnatal day range 30–120; mean age = 52 days) were terminally anesthetized with isoflurane, and once unresponsive, were transcardially perfused with ice-cold sucrose-ACSF containing the following (in mM): 87 NaCl, 75 sucrose, 2.5 KCl, 26 NaHCO$_3$, 1.25 NaH$_2$PO$_4$, 10 glucose, 1.0 CaCl$_2$, and 2.0 MgCl$_2$, saturated with 95% O$_2$/5% CO$_2$. Next, the mice were decapitated and the brains were extracted. The caudal part of the brain was glued to a stage, such that the rostral part of the brain was pitched forward 15$^0$. The stage was then placed into a chamber filled with bubbled ice-cold sucrose ACSF, and the brain was sliced into 300 µm-thick coronal sections using a Leica VT1200S vibratome. The slices were incubated at 35 °C for 30 min in the above sucrose solution or the recording-ACSF described below and then transferred to room temperature for at least 1 hr before recording commenced.

## Electrophysiology recordings

Slices were transferred to a recording chamber perfused with ACSF containing (in mM): 120 NaCl, 2.5 KCl, 25 NaHCO$_3$, 1.4 NaH$_2$PO$_4$, 21 glucose, 0.4 Na-Ascorbate, 2 Na-Pyruvate, 2 CaCl$_2$, and 1 MgCl$_2$, saturated with 95% O$_2$/5% CO$_2$ and maintained at 29–32°C. For some experiments, the bath solution contained the NMDA receptor blocker D-AP5 (25 µM; Abcam), the AMPA receptor blocker CNQX (10 µM; Abcam), or the GABA$_B$ receptor blocker CGP-35348 (60 µM; Tocris). Neurons were visualized on an upright Olympus microscope (BX50WI or BX51WI) using DIC and fluorescence illumination from an LED power source (Mightex) for tdTomato and/or GFP. All neurons were recorded in the barrel field of the primary somatosensory cortex in layers 2–5. Layers were identified visually under DIC optics using several features. The L1-2 border was marked by a sharp increase in soma density from L1. The L3-4 border was identified under 4 x power by the top of the barrels, the presence of horizontal fibers under 40 x (in L4), and the absence of pyramidal cells (PCs) seen in L2/3. L5a was a clear band below L4 and the L5b-6 border was determined under 40 x by a sharp drop in PCs. Neurons were recorded in whole-neuron patch-clamp in current clamp and voltage clamp mode using an internal solution containing (in mM): 130 K-Gluconate, 10 HEPES, 1.1 EGTA, 2 Mg-ATP, 0.4 Na-GTP, 10 Na-phosphocreatine, and 1.5 MgCl$_2$, 0.3–0.5% biocytin, and titrated with 1 M KOH to a final pH of 7.3. Glass pipettes were pulled on a horizontal puller (Sutter Instruments) using borosilicate glass (inner/outer diameter 1.5 mm / 0.86 mm) and had resistances of 2–6 MΩ. Before gaining whole-neuron access, a gigaseal was obtained, and the pipette capacitance was compensated. Access resistances were monitored throughout recordings and were completely compensated. The intrinsic properties of neurons with access resistances >40 MΩ were not analyzed. All data were collected using a Multiclamp 700B amplifier (Molecular Devices), a Digidata digitizer (1440 A or 1550B series, Molecular Devices), and Clampex version (10.6 or 10.7) software (Molecular Devices); data were sampled at 20 kHz and low-pass filtered at 10 kHz.

## Electrophysiological characterization and analysis

A total of 178 cells were recorded. This includes a sample of 76 cells that were randomly selected for patching, and this group was used to determine the proportions of each type. In addition, neurons were recorded in the same Id2-CreER; Dlx5/6-Flpe; Ai65 mice but recordings were biased to superficial layer 2/3 to maximize the chances of obtaining non-LS Id2 INs. Lastly, a third group included cells from animals where NPY cells were labeled with GFP (i.e., Id2-CreER; Dlx5/6-Flpe; Ai65; NPY-hrGFP) to facilitate the identification of non-LS Id2 INs. Neurons were characterized for intrinsic and active properties in current-clamp mode from a resting potential of ~−70 mV. Neurons were injected with 1 s long square pulses of increasing current. Parameters relating to the neuron's intrinsic and active properties were assessed as follows:

### First spike latency and cell type classification

Once rheobase was determined, several current steps were delivered at rheobase to characterize the behavior of the cell. The recorded cells fell into three readily separable groups. The first and most numerous group exhibited a single-spike at rheobase with latencies of >500ms and were termed late-spiking (LS). The second group displayed a high-frequency (>100 Hz) burst of two or more APs at rheobase and were termed burst-spiking (BS). The third group of cells had single spikes at rheobase with short latencies (<100ms) consistent with the charging of the membrane capacitance (first AP latency normalized to membrane $\tau$, mean ± SEM: 4.3±0.5 compared to LS: 70.1±4.1). These cells also exhibited irregular spiking (CV of ISI: 0.112±0.029 vs 0.053±0.004 in LS, p=0.0066, Mann-Whitney Test, see the section below for how CV of ISI was measured) and were classified as irregular-spiking (IS).

### Input resistance

This was determined from Ohm's Law based on the voltage response to a 150ms long negative voltage step of typically 20 pA.

### Membrane time constant ($\tau$)

This was determined by fitting an exponential decay to the voltage response from 1 s long current pulses below rheobase.

### AP waveform properties

To measure these properties, only the first APs from each sweep were considered, as APs in neurons with initial bursts tended to widen significantly for subsequent APs, and all such APs from sweeps with total AP counts <50 were considered. AP threshold and peak were calculated first and then t=0 was defined at the time of AP peak. AP threshold was the voltage reading when the voltage derivative was 20 mV/ms. AP peak amplitude was the maximum voltage reached in the 2ms after the threshold was passed. AP half-width was the time of the AP when the voltage was above the midpoint between the threshold and peak. The afterhyperpolarization (AHP) was calculated as the difference in voltage between the AP threshold and AP reset (minimum voltage in 10ms window after AP peak). Finally, AP max and decay slopes were the maximum and minimum values for dV/dt reached in the 10ms after the threshold was passed. Parameters were calculated separately for each AP and then averaged together for the mean neuronal value.

### Rheobase and firing rates

Rheobase was determined to be the current step that resulted in a single spike, or for bursting cells, a single burst. The firing rate was measured for each voltage sweep by counting the number of spikes in the sweep and taking or interpolating the value at 2 x rheobase.

### Spike frequency adaptation

For LS cells, spike frequency adaptation was measured at 2 x rheobase and near rheobase, during the first depolarization where a spike train was observed throughout the pulse (1.3 x rheobase). For IS and BS cells, the adaptation index was only measured at 2 x rheobase, since near rheobase, firing was often highly irregular. The adaptation index was measured as follows: For each voltage sweep during

which the neuron spiked at least six times (five ISIs), an exponential function, $f_{adap}(t)$, was fit to the firing-rate vs. time plot. The adaptation index (AI) for that sweep was defined as AI = $f_{adap}$(1000ms) / $f_{adap}$(100ms), and the values at 1.3 x rheobase or 2 x rheobase were taken or interpolated to represent adaptation for the cell.

### Spiking regularity

For each voltage sweep, all inter-spike intervals (ISIs) from the last 500ms of the current step were considered. The spiking regularity was measured as the coefficient of variation (CV) for these ISIs, and the reported value was taken from the sweeps at 2 x rheobase.

## DSI – optogenetics and paired recordings

For optogenetics experiments, L2/3 PCs were identified visually under DIC, patched in whole-cell mode, and then confirmed as PCs by a brief electrophysiological characterization. Blue light (470 nm) pulses (2 ms />100 mA) were delivered to the slice using TTL-pulses to a Mightex brand LED under 40 x power with the PC soma centered. Optically evoked IPSCs were recorded under a voltage clamp at a holding potential of –50 mV with D-AP5 and CNQX present to block glutamatergic transmission and CGP 35348 to block the large GABA$_B$ component of the current characteristic of NGFCs (*Oláh et al., 2009*). Cannabinoid dependence of DSI was assessed by optogenetic experiments carried out in the presence of the CB1 receptor blocker AM251 (10 µM; Abcam).

For paired recordings, Id2 neurons were identified as tdT +soma in the Id2-CreER; Dlx5/6-Flpe; Ai65 mouse line or tdT+/GFP ± somas in the Id2-CreER; Dlx5/6-Flpe; Ai65; NPY-hrGFP cross. The Id2 firing pattern was briefly characterized, and the neuron was tagged as late-spiking (LS), irregular-spiking (IS), or burst-spiking (BS). A nearby PC was also patched (Id2-PC inter-soma distance range 10–130 µm, average distance ± SEM = 53±4), and connectivity to the Id2 cell was assessed under voltage-clamp for both cells by eliciting escape spikes from the Id2 cell and looking for the presence of IPSCs in the PC at a holding voltage of –50 mV. If a connection was not seen, either a new Id2-PC pair was tested, or the PC pipette was withdrawn with the Id2 neuron still held, and another nearby PC was patched and tested. If a connection was identified, then the DSI protocol was run as described in *Figure 4*.

## Morphology

During whole-cell electrophysiological recordings, neurons were held for at least 15 min with an internal solution containing ~0.4 or~1.0% biocytin. Some slices, especially the ones with neurons held close to 15 min, were retransferred to the incubation chamber to allow for more biocytin diffusion inside the recorded cell. After filling, the slices were fixed with 4% paraformaldehyde/PBS (diluted from a 32% stock; Electron Microscopy Sciences cat#15714) for 1–7 days at 4 °C. Fixed slices were then thoroughly washed with PBS and left overnight at 4 °C in a 0.4% streptavidin (AlexaFluor 647 conjugate; Invitrogen) solution (498 µl 0.3% Triton X-100 in PBS, 2 µl streptavidin per slice). Slices were then washed in PBS, mounted with Fluoromount-G (Invitrogen) on a glass microscope slide, and imaged under a 40–63 x oil-immersion objective using a confocal microscope (Zeiss). Confocal images were then used to morphologically reconstruct neurons in three dimensions using Neurolucida. In a few cases, cells were juxtacellularly labeled and processed using CUBIC before morphological reconstruction (*Schuman et al., 2019*).

## Silicon probe implantation and recordings

Mice (n=5, 28–35 g, 3–10 months old) were implanted with 64-site silicon probes (NeuroNexus, Cambridge NeuroTech or Diagnostic Biochips) in the PPC (AP 2.0 mm, ML 1.75 mm, DL 0.6 mm), as described previously (*Valero et al., 2021*). Ground and reference wires were implanted in the skull above the cerebellum, and a grounded copper mesh hat was constructed, protecting, and electrically shielding, the probes. Probes were mounted on plastic microdrives that were advanced to layer 6 over the course of 5–8 d after surgery. A 100–200 µm fiber optic was attached to one of the shanks of the silicon probe. After implantation, animals were allowed to recover for at least 1 week and were housed individually under standard conditions (71–73 °F and 40–50% relative humidity) in the animal facility and kept on a 12 h reverse light/dark cycle. We recorded the mice while they slept or walked around freely in the home cage, and the recording session started 1–2 hr after the onset of the dark phase.

Electrophysiological data were acquired using an Intan RHD2000 system (Intan Technologies LLC) digitized with a 30 kHz rate. The wide-band signal was downsampled to 1.25 kHz and used as the LFP signal. For optogenetic tagging of specific neuron types, blue laser light (450 nm, Osram) pulses were delivered in the PPC/V1. The maximum light power at the tip of the optic fiber was 1–4 mW. 100 ms light pulses with 70% of the maximum power were delivered (n=500–1000 times at each intensity at 400±200 ms random intervals).

## Unit activity analysis

Spike sorting was performed semi-automatically with KiloSort 1 (https://github.com/cortex-lab/KiloSort; RRID:SCR_016422), as previously described (*Valero et al., 2021*) and using our own pipeline KilosortWrapper (a wrapper for KiloSort, DOI; https://github.com/brendonw1/KilosortWrapper; *Peterson et al., 2020*). This was followed by manual adjustment of the waveform clusters using the software Phy 2 (https://github.com/kwikteam/phy; *Cyrille, 2021*) and plugins for phy designed in the laboratory (https://github.com/petersenpeter/phy-plugins; *Petersen, 2019*). Unit clustering generated three clearly separable groups based on their spike autocorrelograms, waveform characteristics and firing rate (*Petersen et al., 2021*). Pyramidal cell (PC), narrow waveform (NW) INs and wide waveform interneurons (WW) were tentatively separated based on these two clusters. A more reliable cell identity was assigned after inspection of all features, assisted by monosynaptic excitatory and inhibitory interactions between simultaneously recorded, well-isolated units and light responses (*Valero et al., 2021*; *Valero et al., 2022*). Units were defined as optogenetically responsive cells based on the combination of three criteria (https://github.com/valegarman/hippoCookBook; *Manuel and Abad , 2022*): (*Garcia-Munoz and Arbuthnott, 2015*) an average firing response higher than 2SD, (*D'Souza and Burkhalter, 2017*) significant modulation using a p-value cutoff of $10^{-3}$ (*Lim et al., 2018*) and (*Ibrahim et al., 2020*) against randomly shuffled pulse times (500 replicates) and testing for significant difference between the observed value and the random distribution. To calculate the layer identity of the units, we aligned depth profiles of electrophysiological landmarks. These included the largest amplitude peak of high-frequency LFP power (500 Hz–5 kHz) corresponding to mid-layer 5 and four prominent sinks and sources from the averaged down–up current-source density (CSD) maps for each animal, as previously described (*Valero et al., 2021*).

Rank-order estimation was used to identify the position of units repeating consistently across up events. The timing from multiple units recorded simultaneously was transformed in a normalized sequence order from 0 to 1 within an up event. The rate centroid from each unit in an up event was considered for the rank order.

Brain state scoring was performed as previously described (*Valero et al., 2021*; *Valero et al., 2022*; *Watson et al., 2016*) with the MATLAB resource SleepScoreMaster (https://github.com/valegarman/hippoCookBook). Briefly, spectrograms were constructed with a 1 s sliding 10 s window FFT of 1250 Hz data at log-spaced frequencies between 1 and 100 Hz. Three types of signals were used for state scoring: broadband LFP, narrowband theta frequency LFP, and electromyogram (EMG). For broadband LFP signal, principal components analysis (PCA) was applied to the z-transformed (1–100 Hz) spectrogram. The first principal components in all cases were based on power in the low (<20 Hz) frequency range and had negatively weighted power in the higher (>32 Hz) frequencies. The state scoring algorithm was performed by a series of divisions with thresholds set at the trough between the peaks of distributions in these three metrics. After automated brain state scoring, all states were manually reviewed by the experimenter. The awake-sleep index was estimated as ($rate^{Awake}$ - $rate^{Sleep}$)/ ($rate^{Awake}$ +$rate^{Sleep}$), where $rate^{Awake}$ is the average between the firing rate during Run and Quiet, and $rate^{Sleep}$ was the average between REM sleep and NREM sleep.

## Statistical analysis

Statistical analyses were performed with standard MATLAB functions or GraphPad Prism. No specific analysis was used to estimate a minimal population sample, but the number of animals, trials, in vitro and in vivo recorded cells were larger or similar to those employed in previous works (*Valero et al., 2022*). All data presented were obtained from experimental replicates with at least four independent experimental repeats for each assay. All attempts of replication were successful. Data collection was not performed blinded to the subject conditions. Data analysis was performed blinded to the scorer or did not require manual scoring. Unless otherwise noted, for all tests, non-parametric

two-tailed Wilcoxon's paired signed-rank test and Kruskal-Wallis one-way analysis of variance were used. For multiple comparisons, Tukey's honestly significant difference (HSD) test was employed, and the corrected *p<0.05, **p<0.01, ***p<0.001 are indicated. p-Values for Spearman's correlations are computed using a Student's t distribution for a transformation of the correlation. Results are displayed as median +/-25th/75th percentiles, unless indicated otherwise. Dispersion represents ± IC95.

## Acknowledgements

This work was supported by NIH grants P01NS074972 and R01NS110079 (PI: BR), and U19NS107616 (PI: GB).

## Additional information

### Funding

| Funder | Grant reference number | Author |
| --- | --- | --- |
| National Institutes of Health | P01NS074972 | Bernardo Rudy |
| National Institutes of Health | R01NS110079 | Bernardo Rudy |
| National Institutes of Health | U19NS107616 | György Buzsáki |

The funders had no role in study design, data collection and interpretation, or the decision to submit the work for publication.

### Author contributions

Robert Machold, Conceptualization, Data curation, Supervision, Investigation, Visualization, Methodology, Writing – original draft, Writing – review and editing; Shlomo Dellal, Formal analysis, Investigation, Visualization, Methodology, Writing – original draft, Writing – review and editing; Manuel Valero, Data curation, Software, Formal analysis, Investigation, Visualization, Methodology, Writing – original draft, Writing – review and editing; Hector Zurita, Data curation, Investigation, Visualization, Writing – review and editing; Ilya Kruglikov, Benjamin Schuman, Investigation; John Hongyu Meng, Formal analysis; Jessica L Hanson, Yoshiko Hashikawa, Investigation, Visualization; György Buzsáki, Resources, Funding acquisition; Bernardo Rudy, Conceptualization, Resources, Supervision, Funding acquisition, Writing – review and editing

### Author ORCIDs

Robert Machold ⓘ http://orcid.org/0000-0002-6261-496X
György Buzsáki ⓘ http://orcid.org/0000-0002-3100-4800
Bernardo Rudy ⓘ http://orcid.org/0000-0003-1367-7136

### Ethics

All experimental animals were handled with care to minimize suffering in accordance with institutional animal care and use committee (IACUC) protocols approved by the Division of Comparative Medicine at the NYU Langone Medical Center for Dr. Bernardo Rudy's lab (#IA15-01465 and #IA15-01473).

### Decision letter and Author response

Decision letter https://doi.org/10.7554/eLife.85893.sa1
Author response https://doi.org/10.7554/eLife.85893.sa2

## Additional files

### Supplementary files

• MDAR checklist

## Data availability

All data generated or analyzed during this study are included in the manuscript and supporting files. Source data files have been provided for Figures 1, 2, 4, Table 1, Figure 1—figure supplement 1 and Figure 1—figure supplement 2.

The following previously published dataset was used:

| Author(s) | Year | Dataset title | Dataset URL | Database and Identifier |
|---|---|---|---|---|
| Yao Z, van Velthoven CTJ, Nguyen TN, Goldy J, Sedeno-Cortes AE, Baftizadeh F, Bertagnolli D, Casper T, Chiang M, Crichton K, Ding S, Fong O, Garren E, Glandon A, Gouwens NW, Gray J, Graybuck LT, Hawrylycz MJ, Hirschstein D, Kroll M, Lathia K, Lee C, Levi B, McMillen D, Mok S, Pham T, Ren Q, Rimorin C, Shapovalova N, Sulc J, Sunkin SM, Tieu M, Torkelson A, Tung H, Ward K, Dee N, Smith KA, Tasic B, Zeng H | 2021 | Mouse Whole Cortex and Hippocampus 10x | https://portal.brain-map.org/atlases-and-data/rnaseq/mouse-whole-cortex-and-hippocampus-10x | Allen Brain Map, mouse-whole-cortex-and-hippocampus-10x |

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
