## [Editor Report]

The authors provide a detailed and convincing characterization of a diverse group of cortical interneurons that express the marker Id2, and so can be labeled in Id2-creERT2 mice, but which do not express markers for the previously identified main groups of interneurons: Pvalb, Sst and VIP. This multimodal in vivo and in vitro characterization will be extremely valuable to neuroscientists wishing to study this group of interneurons, which includes previously identified neurogliaform cells and CCK basket cells. in vivo results suggest that these neurons show rebound firing after being inhibited during NREM sleep.

---

## [Decision Letter]

**Decision letter after peer review:**

Thank you for submitting your article "Id2 GABAergic interneurons: a neglected fourth major group of cortical inhibitory cells" for consideration by *eLife*. Your article has been reviewed by 3 peer reviewers, and the evaluation has been overseen by Sacha Nelson as the Reviewing Editor (Reviewer #1) and John Huguenard as the Senior Editor. The following individuals involved in the review of your submission also agreed to reveal their identity: SRamesh Chittajallu (Reviewer #2); Ariel Agmon (Reviewer #3).

Essential revisions:

Each of the reviewers has made multiple suggestions for clarification and or additional analyses likely to improve the impact of the paper. All of these suggestions are included below. Many of the points can be addressed with textual or figure changes and nearly all should be addressable without requiring additional experiments. Additional experiments, while recommended, are not deemed essential.

*Reviewer #1 (Recommendations for the authors):*

The idea that activation of these neurons "…might drive a widespread switch" (line 381) is a bit hard to square with what I presume are rather modest changes in firing rate. The changes in firing rates following circuit stimulation should be reported in Hz. (or in Hz and SD). The translation between Hz and SD is difficult to assess, but the fact that the changes in rate are only from ~3 to ~2 from wake to sleep, suggests that the changes following stimulation correspond to relatively small changes in rate. The reason this is important to assess is that the synaptic output of these neurons and hence their impact on the circuit is likely dependent on their rate of firing. The idea of a "widespread switch" implies a very large impact. But it is not clear this is supported. I find it significant that any impact can be demonstrated following the stimulation of a modest number of interneurons, so I do not think it is necessary to oversell this. On the other hand, if the authors really do think the data support the "widespread switch" model, the reader could use some more convincing.

*Reviewer #2 (Recommendations for the authors):*

1. Figure S1 is rather confusing in its layout particularly since the reference to it in the manuscript is limited to two very brief sentences. It may be helpful for clarity if the authors expand a little in the text to explain the data distilled in this supplemental figure since, along with Figure 1, it sets the rationale for the genetic strategy employed.

2. Although the main focus, in this manuscript is L2-L5 it would be useful in the context of the authors' previous publication to mention perhaps in the discussion whether the layer 1 reported cells using the current strategy labels canopy cells as well.

3. Lns 346-349. It appears that a substantial number of INs are reported in Layer 6 of the Id2Cre:Dlx5/6flp (Figure 1C) when compared to that in Id2Cre:Nkx2-1Flp mouse. Since approximately 60% (Figure 2) are NPY this suggests there are significant CGE NFCS in this deep layer in addition to the sparse number of MGE-derived ones previously investigated. At the very least, an example of their morphology and basic characterization of their firing properties should be included to add to Figure 3 and Table 1. In light of their previous study (Valero et al., 2021), the authors should briefly discuss whether they suspect CGE vs MGE ID2 cells in layer 6 to perform similar functional roles and respond in a similar manner during network/behavioral states (i.e. embryonic origin redundant)?

4. The focus of Figures 3, 4 and Table 1 is to demonstrate morpho-functional differences between the LS and non-LS cells reported in Id2cre:Dlx5/6Flp mouse line. Although the characterization of intrinsic properties is well performed, the evaluation of the synaptic properties of LS and non-LS ID2 INs would benefit from some additional data. This figure could be fleshed out further as follows:

(a) Throughout the manuscript the authors point out the potentially widespread influence of NGFCs (e.g. ln 74-75) which is a consequence of the well-characterized volume transmission. However, they do not include the percentage success of attaining a paired recording between LS and non-LS ID2 cells and PCs – this would not require further experimentation since paired recordings have already been performed.

(b) In addition, it is well known that the decay of the GABAA IPSC is markedly prolonged when these receptors are activated by NGFCs – therefore a plot of this parameter demonstrating a clear contrast between LS and non-LS ID2 should be included. Again, this should not require further experimentation.

(c) The authors could also include sample traces of the pharmacologically isolated large GABAB-evoked current that is reminiscent of NGFCs (via optogenetic stimulation and/or paired recordings).

(d) Finally, the authors, use DSI as a measure of CB1 expression to demonstrate a divergence in the NGFC and CCK populations. This is a rather circuitous route to probe for functional CB1 receptors and a negative effect may not necessarily preclude functional CB1 receptor expression. A more direct method would be to perform ChR2 and paired recordings and assay the effects of CB1 receptor agonist (WIN55,212-2) on LS and non-LS ID2 cells mediated IPSC amplitude.

(e) Additional data to further highlight the advantage of using the Id2Cre:DLX5/6Flp mouse to increase the ease to which cortical NGFCs (particular in deeper layers) can be targeted to assess other aspects of their synaptic properties. Perhaps an aspect that has not been described/studied previously would be welcome.

*Reviewer #3 (Recommendations for the authors):*

My comments are mostly recommendations to the author on using more precise language, enhancing their analysis, tightening the discussion of their results and conclusions, and adding citations to a few overlooked but important previous studies of NGFCs.

1. Please tone down a bit the description of NGFCs as "neglected", "enigmatic", or "mysterious" in the Title, Abstract, Introduction and Discussion.

2. The first paragraph in Results should be incorporated into the early part of the Introduction. The introduction needs to mention 5HT3aR neurons as a group that encompasses all (or most) CGE-derived interneurons (as described in previous work by the authors and others), and further subdivided into VIP- and non-VIP-expressing groups, the latter now being labeled by the new intersectional strategy.

3. Line 31: some references are expected at the end of this heavily loaded sentence.

4. Lines 50, 58: Although not explained, I assume that the 12% and 18% numbers come from the authors' previous estimate that VIP cells account for about 40% of 5HT3a cells, which in turn account for about 30% of all interneurons. However the product of two approximate numbers is still an approximate number, so it is important to explain here how the 12 and 18% estimates were derived lest readers take these numbers at face value. That these estimates were confirmed in the current study can be indicated later, in the Results and Discussion.

5. Line 90: Please add a citation of Vucurovic, K., Gallopin, T., Ferezou, I., Rancillac, A., Chameau, P., van Hooft, J. A., Geoffroy, H., Monyer, H., Rossier, J., and Vitalis, T. (2010).

6. Line 114: I assume that "well-labeled brains" refers to the more successful induction of CreER, as explained later in the Discussion; however this term is never defined. Somewhere in the manuscript the criteria for determining what is a "well-labeled brain" need to be enunciated.

7. Line 147: The reference to Schuman et al. is out of place here, because that study never mentions Id2. Thus, another justification or citation is needed here for the implicit assumption that most L1 neurons are Id2 cells.

8. Line 158-159: Do the authors have direct histological evidence, from their own or previous work, that NPY and CCK-expressing cells are non-overlapping populations? This evidence should be presented or cited here.

9. Line 160: It may be worthwhile pointing out here that other than L1 and L6, the fractions of NPY+ and CCK^+^ Id2 cells (Figure 2C) sum up to 100{plus minus}5%, so these two groups account for all or nearly all Id2 cells in these layers.

10. Line 161-2: Please rephrase this sentence more precisely. For example, "Our in-situ hybridization analysis (Figure 2A-B) suggested that the majority (87%) of Id2 cells in L2-5 are NPY-expressing NGFCs."

11. Lines 161-176, 187-194, and Figure 3: I recommend that the authors expand their electrophysiological analysis. As currently written, the distinction between the different electrophysiological phenotypes is rather subjective, and a clear quantitative definition of these phenotypes is missing. Some of the electrophysiological parameters in Table 1 could be analyzed and plotted in ways that will confirm quantitatively, or at least demonstrate graphically, that LS, BS, and IS are indeed separate groups with distinct properties.

12. Lines 187-194: It makes more sense to move this section immediately after line 176, thus directly juxtaposing the electrophysiological properties of LS and CCK cells. The corresponding morphologies can then be described in a separate paragraph.

13. Lines 199, 209, 226, other: No need to put CCK basket cells in parenthesis.

14. Line 206: Endocannabinoids are NOT transported retrogradely to the presynaptic terminal – they signal retrogradely by binding to the presynaptic terminal from its extracellular side.

15. Line 297: This sentence is incorrect on three counts. The use of neuropeptides, calcium-binding proteins, and other protein markers has been around for at least 4 decades and has not "evolved tremendously over the last decade". It was initiated by Celio and Heizmann in 1981, not by Kawaguchi and Kubota in 1993. Transcriptomic profiling of neurons is a different beast altogether and should not be confounded with protein markers.

16. Lines 315-325: This section contradicts what the authors say in lines 305-315. Based on the (correct) logic of lines 305-315, the Htr3-EGFP line should label LESS CGE-derived neurons than the Htr3-Cre line, not more. Thus it needs to be explained why the authors observe the opposite (Figure S7). It is also not clear how these results are consistent with an early broad expression of Htr3a which later becomes more restricted. If the adult pattern was more restricted than the developmental pattern, then labeling in the Htr3a-EGFP line should be more restricted than in the Htr3a-Flp line, as the expression of EGFP should reflect the adult pattern while the expression of the driver line should be cumulative. Altogether, there is a paradoxical finding here, and it may be better to remove this section (and Figure S7) altogether until the authors figure out what is behind these results.

17. Line 325: Reference 81 appears not to belong here.

18. Line 334: See comment #6 on "well-labeled brains".

19. Line 341: "a variety of CCK^+^ cells" is a very loose term. Maybe the authors mean to say "CCK expressing interneurons with diverse firing patterns".

20. Line 370: Please add a citation to Hay, Y. A., Deperrois, N., Fuchsberger, T., Quarrell, T. M., Koerling, A. L., and Paulsen, O. (2021).

21. Line 435: To describe the range of ages used, please indicate SD and not SEM.

22. Line 437: "perfused transcardially", presumably.

23. Line 441: It is not clear what "angled towards the blade" means; was the plane of section sloping caudally (from front to back) or sloping rostrally (from back to front)?

24. Line 491: The phrase in parenthesis is garbled.

25. Figure 2: Please be consistent with the text labeling of the different panels. E.g., panels A-B are labeled by fluorescent protein (tdTomato) while panels C-D are labeled by the gene (Id2), even though all panels are from the same genotype.

26. Legend to Figure 3: The connection probabilities for LS-PC and nonLS-PC are important data; please report them in the body of the manuscript.

27. Legend to Figure 5: Designate p-values by lowercase p.

28. Figure 5I: The negative correlation of the Id2 cells seems to rely on a single outlier and is therefore not convincing.

---

## [Author Response]

Essential revisions:Each of the reviewers has made multiple suggestions for clarification and or additional analyses likely to improve the impact of the paper. All of these suggestions are included below. Many of the points can be addressed with textual or figure changes and nearly all should be addressable without requiring additional experiments. Additional experiments, while recommended, are not deemed essential.Reviewer #1 (Recommendations for the authors):The idea that activation of these neurons "…might drive a widespread switch" (line 381) is a bit hard to square with what I presume are rather modest changes in firing rate. The changes in firing rates following circuit stimulation should be reported in Hz. (or in Hz and SD). The translation between Hz and SD is difficult to assess, but the fact that the changes in rate are only from ~3 to ~2 from wake to sleep, suggests that the changes following stimulation correspond to relatively small changes in rate. The reason this is important to assess is that the synaptic output of these neurons and hence their impact on the circuit is likely dependent on their rate of firing. The idea of a "widespread switch" implies a very large impact. But it is not clear this is supported. I find it significant that any impact can be demonstrated following the stimulation of a modest number of interneurons, so I do not think it is necessary to oversell this. On the other hand, if the authors really do think the data support the "widespread switch" model, the reader could use some more convincing.

We agree with the reviewer that the phrase “widespread switch” was not justified based on the available in vivo data. While it is significant that such a large percentage of putative excitatory cells are inhibited, particularly in L2/3 where NGFCs are most abundant, we don’t know how this compares with stimulating other INs such as PV or SST cells. We have now therefore removed this text, with the new version as follows: “We found that these putative NGFCs have a number of unique properties, including relatively constant firing rates throughout different brain states and a strong rebound in activity immediately following the down state during NREM sleep. We also found that optogenetic stimulation of Id2 INs produced widespread inhibition of PCs and other INs, with roughly half of all cells across cortical layers exhibiting a decrease in activity. The notable rebound in NGFC activity following the down state, together with the extensive output connectivity of these neurons, suggests an important role for this form of GABAergic inhibition in setting the stage for new incoming information during bouts of cortical activity”. (lines 420-428)

In addition, based on the reviewer suggestion we have now added an additional panel in Figure 6 (6C) that shows the changes in firing rates in Hz to complement Figure 6B (changes in firing rates shown as changes in SD). It is important to note that the overall circuit effects of Id2 INs (or other cells) stimulation are complex, with both inhibition and disinhibition observed over time. In both panels, the overall (mean) effects are shown for each class of cells (NW, WW, Pyr), along with the effects on subsets of the latter separated into quintiles based on response.

Reviewer #2 (Recommendations for the authors):1. Figure S1 is rather confusing in its layout particularly since the reference to it in the manuscript is limited to two very brief sentences. It may be helpful for clarity if the authors expand a little in the text to explain the data distilled in this supplemental figure since, along with Figure 1, it sets the rationale for the genetic strategy employed.

We agree with the reviewer and have extensively revised Figure S1 (now Figure 1 —figure supplement 1; see response to Reviewer 1 above).

2. Although the main focus, in this manuscript is L2-L5 it would be useful in the context of the authors' previous publication to mention perhaps in the discussion whether the layer 1 reported cells using the current strategy labels canopy cells as well.

Yes, based on the overall labeling in L1 (83%), our Id2 IN targeting strategy does label canopy cells as well as a7 cells. In our previous L1 work (Schuman et al., 2019) we estimated that ~90% of INs in L1 were non-VIP, which corresponds fairly well with the 83% labeling observed using Id2-CreER here (Figure 1E), particularly when taking into account some variability in labeling efficacy across Id2 IN subtypes. For example, while we could not assess canopy cells directly, we did observe that ~60% of the a7 cells in L1 (stained with fluorescent α-bungarotoxin as described in Schuman et al., 2019) were labeled in the Id2-CreER; Dlx5/6-Flpe; Ai65 animals.

The following text has been added to the Discussion: “Using our Id2 intersectional genetic strategy, we find that 83% of L1 INs are labeled (Figure 1E), close to the expected 90% of non-PV/non-SST/non-VIP INs, suggesting that all non-VIP INs in L1 are Id2 cells.” (lines 374-377)

3. Lns 346-349. It appears that a substantial number of INs are reported in Layer 6 of the Id2Cre:Dlx5/6flp (Figure 1C) when compared to that in Id2Cre:Nkx2-1Flp mouse. Since approximately 60% (Figure 2) are NPY this suggests there are significant CGE NFCS in this deep layer in addition to the sparse number of MGE-derived ones previously investigated. At the very least, an example of their morphology and basic characterization of their firing properties should be included to add to Figure 3 and Table 1. In light of their previous study (Valero et al., 2021), the authors should briefly discuss whether they suspect CGE vs MGE ID2 cells in layer 6 to perform similar functional roles and respond in a similar manner during network/behavioral states (i.e. embryonic origin redundant)?

We agree with the reviewer that a comparison of CGE vs MGE derived NGFC in L6 would be interesting; however, this is challenging at present since there is no straightforward genetic strategy to selectively label CGE derived NGFC. We did perform a comparison of electrophysiological properties between MGE NGFC (Id2/Nkx2.1) and L1 NGFC (NDNF/NPY; CGE derived) in Valero et al. 2021 which indicated that the two NGFC populations were largely similar. Based on the results from Valero et al. which showed that the sparse MGE NGFC in L6 comprise the majority of down state active cells, it appears that CGE NGFC in L6 (a much larger population than the MGE NGFC) are not down state active cells and thus behave in a distinct manner in vivo. Furthermore, we did not observe down-state active cells in our recordings of deep layer Id2 INs (e.g. Figure 5G).

We have added the following text to the Discussion:

“Within the Id2 NGFC in L6, a subset originates from the MGE (i.e., Nkx2.1 lineage) and exhibit a striking anti-correlated activity pattern, with peak activity during cortical down states (60). We did not observe down-state active cells in our Id2 IN recordings (Figure 5G), indicating that NGFC of CGE origin (the majority of cortical NGFC) behave in a distinct manner in vivo. Thus, like L1, L6 is a specialized case that merits its own study to elucidate the diverse IN species that reside there.” (lines 386-392)

4. The focus of Figures 3, 4 and Table 1 is to demonstrate morpho-functional differences between the LS and non-LS cells reported in Id2cre:Dlx5/6Flp mouse line. Although the characterization of intrinsic properties is well performed, the evaluation of the synaptic properties of LS and non-LS ID2 INs would benefit from some additional data. This figure could be fleshed out further as follows:(a) Throughout the manuscript the authors point out the potentially widespread influence of NGFCs (e.g. ln 74-75) which is a consequence of the well-characterized volume transmission. However, they do not include the percentage success of attaining a paired recording between LS and non-LS ID2 cells and PCs – this would not require further experimentation since paired recordings have already been performed.

We have added this information that was previously only in the legend for Figure 4 to the Results: “In paired recordings of L2/3 LS Id2 cells and PCs, we observed a very high connection probability (70%; 19/27 pairs tested), consistent with previous observations and the hypothesis that NGFCs mediate volume transmission of GABA (40).” (lines 183-185)

(b) In addition, it is well known that the decay of the GABAA IPSC is markedly prolonged when these receptors are activated by NGFCs – therefore a plot of this parameter demonstrating a clear contrast between LS and non-LS ID2 should be included. Again, this should not require further experimentation.

We have added the following text to the Results: “Finally, the decay of the IPSC elicited by NGFC is prolonged compared to other IN inputs; consistent with this, we found a significantly longer decay rate (80-20% of peak) for LS-PC pairs compared with non-LS-PC pairs (LS-PC: 51±8 ms, n=6 pairs; non-LS-PC: 16±3 ms, n=5 pairs; p=0.005, two-tailed unpaired t-test).” (lines 186-190)

(c) Finally, the authors, use DSI as a measure of CB1 expression to demonstrate a divergence in the NGFC and CCK populations. This is a rather circuitous route to probe for functional CB1 receptors and a negative effect may not necessarily preclude functional CB1 receptor expression. A more direct method would be to perform ChR2 and paired recordings and assay the effects of CB1 receptor agonist (WIN55,212-2) on LS and non-LS ID2 cells mediated IPSC amplitude.

We agree that additional paired recordings comparing LS and non-LS cell mediated IPSCs would be interesting; however, finding connected non-LS to Pyr pairs for these experiments was quite challenging. In the overall Id2 population, using optogenetics, we did observe DSI and found that this effect was blocked by the CB1R-specific antagonist AM251 (Figure 4).

Reviewer #3 (Recommendations for the authors):My comments are mostly recommendations to the author on using more precise language, enhancing their analysis, tightening the discussion of their results and conclusions, and adding citations to a few overlooked but important previous studies of NGFCs.1. Please tone down a bit the description of NGFCs as "neglected", "enigmatic", or "mysterious" in the Title, Abstract, Introduction and Discussion.

While we replaced most of these descriptors such as enigmatic or mysterious, we left the term “neglected” in the title. First, the term applies to the group, i.e., the INs that are not PV, SST or VIP. The fact that the majority of them are NGFCs is a finding of this study. Second, examination of the scores of papers investigating the role of different IN types in various functions in the last five years shows they all compare PV, SST and VIP subtypes at best, which neglects the remaining ~18% of INs highlighted in this study.

2. The first paragraph in Results should be incorporated into the early part of the Introduction. The introduction needs to mention 5HT3aR neurons as a group that encompasses all (or most) CGE-derived interneurons (as described in previous work by the authors and others), and further subdivided into VIP- and non-VIP-expressing groups, the latter now being labeled by the new intersectional strategy.

We have now moved the first paragraph in the Results to the Introduction as suggested.

3. Line 31: some references are expected at the end of this heavily loaded sentence.

We have added references for several recent reviews on this topic (Schuman et al., 2021 and Aru et al., 2020)

4. Lines 50, 58: Although not explained, I assume that the 12% and 18% numbers come from the authors' previous estimate that VIP cells account for about 40% of 5HT3a cells, which in turn account for about 30% of all interneurons. However the product of two approximate numbers is still an approximate number, so it is important to explain here how the 12 and 18% estimates were derived lest readers take these numbers at face value. That these estimates were confirmed in the current study can be indicated later, in the Results and Discussion.

We provide references for several recent reviews that cover the topic of IN group proportions; these estimates are not just from our own work.

5. Line 90: Please add a citation of Vucurovic, K., Gallopin, T., Ferezou, I., Rancillac, A., Chameau, P., van Hooft, J. A., Geoffroy, H., Monyer, H., Rossier, J., and Vitalis, T. (2010).

We have added this citation as requested.

6. Line 114: I assume that "well-labeled brains" refers to the more successful induction of CreER, as explained later in the Discussion; however this term is never defined. Somewhere in the manuscript the criteria for determining what is a "well-labeled brain" need to be enunciated.

We have added the following sentence to the Methods: *“The efficacy of tamoxifen induction of CreER mediated reporter activation was assessed by histology (see below); well labeled brains typically exhibited >50 labeled cells in a 20 μm thick tissue section of S1BF encompassing 4-5 barrels (~1300 μm x 1700 μm field).” (lines 452-455)*

7. Line 147: The reference to Schuman et al. is out of place here, because that study never mentions Id2. Thus, another justification or citation is needed here for the implicit assumption that most L1 neurons are Id2 cells.

In Schuman et al., we estimated that ~90% of L1 INs were non-PV, SST, or VIP expressing subtypes. Here, using Id2-CreER; Dlx5/6-Flpe; FLTG intersectional/subtractive genetics, we observe labeling of 83% of L1 INs (Figure 1E), which corresponds well with the Schuman et al. estimate.

8. Line 158-159: Do the authors have direct histological evidence, from their own or previous work, that NPY and CCK-expressing cells are non-overlapping populations? This evidence should be presented or cited here.

We did not see any overlap with CCK IHC and NPY-hrGFP in our tissue sections. This is consistent with the Allen transcriptome data. In Figure S1C (now Figure 1 —figure supplement 1, panel C), we show plots of the trimmed mean cpm values for *Npy* and *Cck* mRNA from Yao et al., 2021 across the Id2 populations that are expected to be labeled in our experiments. The main NGFC bin (#12) exhibits the highest levels of *Npy* mRNA as expected, while its mean *Cck* mRNA levels are near zero.

We added the following sentence to the Results: “No overlap was observed between strong CCK IHC and NPY-hrGFP expression, consistent with previous work and the Allen scRNAseq data (Figure 1 —figure supplement 1).” (lines 162-164)

9. Line 160: It may be worthwhile pointing out here that other than L1 and L6, the fractions of NPY+ and CCK^+^ Id2 cells (Figure 2C) sum up to 100{plus minus}5%, so these two groups account for all or nearly all Id2 cells in these layers.

We added the following sentence to the Results: “Thus, a significant fraction of the Id2/non-NPY population in cortical layers outside of L1 appear to be CCK^+^ IN subtypes (~80% in L2/3), such that Id2/NPY and Id2/CCK populations account for nearly all of the Id2 INs in L2-5.” (lines 164-166)

10. Line 161-2: Please rephrase this sentence more precisely. For example, "Our in-situ hybridization analysis (Figure 2A-B) suggested that the majority (87%) of Id2 cells in L2-5 are NPY-expressing NGFCs."

We have replaced the original text with: “Our FISH analysis (Figure 2A-B) suggested that the majority (87%) of Id2 INs in L2-5 are NPY-expressing NGFCs.” (lines 167-169)

11. Lines 161-176, 187-194, and Figure 3: I recommend that the authors expand their electrophysiological analysis. As currently written, the distinction between the different electrophysiological phenotypes is rather subjective, and a clear quantitative definition of these phenotypes is missing. Some of the electrophysiological parameters in Table 1 could be analyzed and plotted in ways that will confirm quantitatively, or at least demonstrate graphically, that LS, BS, and IS are indeed separate groups with distinct properties.

We have added a plot to Figure 3 (panel G) of LS, BS, and IS cells resolved by AP half-width, rheobase, and AP latency.

12. Lines 187-194: It makes more sense to move this section immediately after line 176, thus directly juxtaposing the electrophysiological properties of LS and CCK cells. The corresponding morphologies can then be described in a separate paragraph.

Rather than trying to combine the ephys results for LS and CCK types, we separated the Id2/CCK text to form a new paragraph: “The remaining Id2 cells (20% in L2/3 and 10% in L4-5) had either an irregular or a bursting firing pattern during low to moderate depolarizations….” (line 202 -)

13. Lines 199, 209, 226, other: No need to put CCK basket cells in parenthesis.

We assume that the reviewer meant quotes, and so we have now removed those from the text.

14. Line 206: Endocannabinoids are NOT transported retrogradely to the presynaptic terminal – they signal retrogradely by binding to the presynaptic terminal from its extracellular side.

We have replaced the original text with the following:

“Depolarization of a connected pyramidal cell leads to the release of endocannabinoids from the pyramidal cell that then bind to CB1 receptors located on the presynaptic terminals of the CCK basket cells, leading to suppression of GABA release (78-80)..” (lines 218-221)

15. Line 297: This sentence is incorrect on three counts. The use of neuropeptides, calcium-binding proteins, and other protein markers has been around for at least 4 decades and has not "evolved tremendously over the last decade". It was initiated by Celio and Heizmann in 1981, not by Kawaguchi and Kubota in 1993.

We have replaced the original text with: “The use of molecular markers to classify subpopulations of cortical GABAergic interneurons (INs) has provided deep insights into IN diversity (6, 65, 84, 85), and has expanded in recent years with the advent of single-cell RNA sequencing methods to profile IN transcriptomes (14, 16, 18, 86).” (lines 314-317)

Transcriptomic profiling of neurons is a different beast altogether and should not be confounded with protein markers.

We respectfully disagree with the reviewer that “Transcriptomic profiling of neurons is a different beast altogether and should not be confounded with protein markers.” While the presence of mRNA does not necessary mean that the corresponding protein is expressed, obviously there can be no protein expression without mRNA, so there is indeed a strong correlation between a cell’s transcriptome and proteome (e.g., Wilhelm et al., 2014 PMID 24870543).

16. Lines 315-325: This section contradicts what the authors say in lines 305-315. Based on the (correct) logic of lines 305-315, the Htr3-EGFP line should label LESS CGE-derived neurons than the Htr3-Cre line, not more. Thus it needs to be explained why the authors observe the opposite (Figure S7). It is also not clear how these results are consistent with an early broad expression of Htr3a which later becomes more restricted. If the adult pattern was more restricted than the developmental pattern, then labeling in the Htr3a-EGFP line should be more restricted than in the Htr3a-Flp line, as the expression of EGFP should reflect the adult pattern while the expression of the driver line should be cumulative. Altogether, there is a paradoxical finding here, and it may be better to remove this section (and Figure S7) altogether until the authors figure out what is behind these results.

We agree that the results with the various Htr3a labeling methods are confusing, and have now added additional text in the Discussion to clarify this “paradoxical finding” as it reveals important considerations when using transgenic labeling methods in general: “Use of BAC transgenics has turned out to be more complicated than expected since each transgenic founder harbors its own positional and copy number variation. The Htr3a(BAC)-EGFP founder line was specifically chosen over others for high levels of EGFP expression. In hindsight, this likely contributed to its pan-CGE IN expression pattern, whereas endogenous Htr3a expression appears to be broad but transient in most CGE INs following their specification, being largely restricted to VIP/CCK and Sncg IN subtypes at adult ages. In contrast to the Htr3a(BAC)-EGFP line, the Htr3a(BAC)-Cre founder was selected based on its cumulative recombination pattern with the cre-dependent tdTomato reporter Ai9, and likely has a much lower copy number or genomic insertion location that results in its expression being closer to the endogenous Htr3a gene. The Htr3a-Flpo line is a single copy knock in whose cumulative labeling with the tdTomato reporter Ai65F is less efficient overall than that seen with either BAC transgenic line, most likely due to the lower/transient expression of Flpo in most CGE INs.” (lines 346-359)

If all three lines were single copy knock ins, then the reviewer’s expectations that there should be fewer labeled cells in the Htr3a-EGFP vs. Htr3a-Cre; Ai9 would likely be met; however, this is not the case.

17. Line 341: "a variety of CCK^+^ cells" is a very loose term. Maybe the authors mean to say "CCK expressing interneurons with diverse firing patterns".

We have edited the text as follows: *“In L2-5, we find that Id2 INs are mostly comprised of*

18. Line 370: Please add a citation to Hay, Y. A., Deperrois, N., Fuchsberger, T., Quarrell, T. M., Koerling, A. L., and Paulsen, O. (2021).

We have added this citation.

19. Line 435: To describe the range of ages used, please indicate SD and not SEM.

We have replaced that text with: *“postnatal day range 30-120; mean age = 52 days” (line 482)*

20. Line 441: It is not clear what "angled towards the blade" means; was the plane of section sloping caudally (from front to back) or sloping rostrally (from back to front)?

We have replaced that text with: “The caudal part of the brain was glued to a stage, such that the rostral part of the brain was pitched forward 15⁰. The stage was then placed into a chamber filled with bubbled ice-cold sucrose ACSF, and the brain was sliced into 300 µm-thick coronal sections using a Leica VT1200S vibratome.” (lines 487-490)

21. Line 491: The phrase in parenthesis is garbled.

We have replaced that text with: “The third group of cells had single spikes at rheobase with short latencies (<100 ms) consistent with the charging of the membrane capacitance (first AP latency normalized to membrane τ, mean ± SEM: 4.3 ± 0.5 compared to LS: 70.1 ± 4.1).” (lines 537-540)

22. Figure 2: Please be consistent with the text labeling of the different panels. E.g., panels A-B are labeled by fluorescent protein (tdTomato) while panels C-D are labeled by the gene (Id2), even though all panels are from the same genotype.

We have fixed this inconsistency.

23. Legend to Figure 3: The connection probabilities for LS-PC and nonLS-PC are important data; please report them in the body of the manuscript.

We have now included those values in the main text

24. Legend to Figure 5: Designate p-values by lowercase p.

We have fixed this.